# Biogeochemical functioning of Lake Alaotra (Madagascar): a reset of aquatic carbon sources along the land-ocean aquatic continuum

Vao Fenotiana Razanamahandry[1*], Alberto Vieira Borges[2], Liesa Brosens[3], Cedric Morana[2], Tantely Razafimbelo[4], Tovonarivo Rafolisy[4], Gerard Govers[1] and Steven Bouillon[1*].

[1]Department of Earth and Environmental Sciences, KU Leuven, Leuven, Belgium
[2]University of Liège, Chemical Oceanography Unit, Liège, Belgium
[3]Environmental Modeling Unit, Flemish Institute for Technological Research (VITO), Mol, Belgium
[4]Laboratoire des Radio-Isotopes, University of Antananarivo, Antananarivo, Madagascar
[*]: Corresponding authors, e-mail: vaofenotiana.razanamahandry@kuleuven.be; steven.bouillon@kuleuven.be

## Abstract

Our understanding of the role of tropical lakes in regional carbon budgets remains hampered by a lack of data covering the vast diversity of lake types and settings. Here, we provide a first comprehensive survey of the carbon (C) biogeochemistry of the Lake Alaotra system, a large shallow lake (surface is 200 km$^2$, maximum depth 2 m) surrounded by an extensive floodplain and ricefields located in the highlands of Madagascar. The current landscape in the region is grassland-dominated and dotted by major gullies called "lavaka", which have historically been claimed to lead to high erosion rates, and would thus also mobilize large amounts of soil C. We investigated the seasonal variability of the concentrations and stable isotope ratios of inorganic and organic C pools, as well as a range of other relevant proxies, including physico-chemical parameters, dissolved $CO_2$ and $CH_4$ concentrations, total alkalinity, and chlorophyll a (Chl-a) from spatially distributed sampling and seasonal monitoring of several rivers. While rivers were found to carry high total suspended matter (TSM) loads with a modest particulate organic C (POC) content, the lake itself and its outflow were characterized by much lower TSM values and high %POC (relative contribution of POC to TSM). The POC concentration of the outflow (13.0 ± 7.7 mg L$^{-1}$) was substantially higher than in the inflowing water (1.9 ± 2.1 mg L$^{-1}$), and $\delta^{13}C$ values were also distinct between inflowing water (-24.6 ± 1.8 ‰) and the lake (-26.5 ± 2.1 ‰) or its outflow (-25.2 ± 1.4 ‰). Similarly, the lake outflow was surprisingly rich in dissolved organic carbon (DOC) (9.5 ± 1.4 mg L$^{-1}$) compared to inflowing water (2.6 ± 1.1 mg L$^{-1}$). This indicates that the lake and its surrounding wetlands act as a substantial source of additional organic C which is exported downstream. The $CO_2$ and $CH_4$ concentrations in inflowing and outflowing rivers were substantially higher than in lake waters, and peaked during the rainy season due to lateral inputs from wetlands. However, sources of POC and DOC were uncoupled: $\delta^{13}C$ data indicated that marsh vegetation was the main source of net DOC inputs, while phytoplankton contributed substantially to POC in the lacustrine waters, at least during parts of the sampling period. Indeed, lake suspended matter has relatively low POC/Chl-a ratios (143–564, particularly during the May sampling period), high %POC (10 to 29 %), and $\delta^{13}C$ values (-26.5 ± 2.1 ‰) distinct from those in marsh-derived organic matter. Despite the evidence for phytoplankton production as a contributor to the lake POC pool, the lake acted as a net source of $CO_2$ to the atmosphere, likely due to the high C inputs from the surrounding marshes, and sediment respiration considering the shallow water depth. Nevertheless, the $pCO_2$ levels in the surface waters of the lake were lower than those in the inflowing and outflowing rivers. This reduction is likely due to the combined effects of phytoplankton production, which assimilates $CO_2$ during photosynthesis, and degassing processes. When $CO_2$-supersaturated

riverine water enters the open lake, increased turbulence caused by wind fetch enhances gas exchange with the
atmosphere, allowing $CO_2$ to escape more readily from the water column. The biogeochemical functioning of
Lake Alaotra differs substantially from the large and deeper East African (sub)tropical lakes and was more similar
to lakes surrounded by flooded forests in the Congo River basin, likely due to a combination of its large surface
area and shallow water depth, and the large extent of surrounding wetlands and floodplains. It acts as an abrupt
element in the land-ocean continuum of the catchment, whereby the biogeochemical characteristics of the
Maningory River (i.e., the lake outflow) are strongly determined by processes taking place in Lake Alaotra and
its wetlands, rather than being reflective of characteristics and processes further upstream in the catchment.

## 1 Introduction

Datasets on the biogeochemistry and C cycling along terrestrial-aquatic continuum in tropical environments are still scarce in comparison to the boreal and temperate zone. Lakes have traditionally been characterized as sources of $CO_2$ to the atmosphere (Cole et al., 1994, 2007) sustained by the production of $CO_2$ from degradation of terrestrial organic matter. Accordingly, lakes would then be net heterotrophic systems sustained by external inputs of allochthonous organic matter (DOC and POC) from the surrounding landscape (catchment) (Del Giorgio et al., 1999). However, the impact of external inputs of allochthonous organic matter on the cycling of organic matter partly depends on the size of the system (surface area and depth), with larger and deeper lakes being less heterotrophic (Del Giorgio and Peters, 1994; Staehr et al., 2012). In tropical lakes, aquatic primary production can be intense due to combined year-round favourable light, temperature conditions, and weak water column stratification, favourable to nutrient inputs from deep to surface waters (Lewis, 2010). Morana et al. (2022) showed that African tropical lakes with a low DOC content (non-humic) were net autotrophic leading to low $CO_2$ sources or even sinks of atmospheric $CO_2$ (Borges et al., 2022). Lakes with high DOC content from wetlands were characterized by low primary production, and were strong sources of $CO_2$ to the atmosphere (Borges et al., 2022). In addition, dissolved organic C (DOC) plays an important function in lake ecosystems and regulates the carbon and energy cycle of inland waters (Wetzel, 2003). The study by Morana et al. (2022) showed that in situ primary production in some of the studied lakes could be ~20 times higher than the organic carbon (C) burial in sediments. This indicates that much of the carbon fixed through phytoplankton production is rapidly recycled within the lake system, through processes such as microbial respiration, grazing, and remineralization, rather than being buried in sediments or emitted as $CO_2$ to the atmosphere. This finding challenges the traditional view of lakes as net heterotrophic systems (Del Giorgio et al., 1993; Duarte and Prairie, 2005; Aufdenkampe et al., 2011).

Irrespective of their trophic status, lakes are often highly active areas in terms of organic matter processing and biogeochemical modifications (Sobek et al., 2006; Tranvik et al., 2018). Moreover, during the transit of water in the lakes, repartitioning of organic matter between dissolved and particulate organic C might take place, changes in characteristics in the lake could occur and might result in a difference between inflow and outflow characteristics (Tranvik et al., 2009; Hanson et al., 2011). While the amount of data on the origin of DOC and POC, and on transport fluxes in tropical rivers (e.g., Cuevas-Lara et al., 2021; Richey et al., 2022), has grown steadily, much less is known about the biogeochemical cycling and OC source contributions in tropical lakes. Global data syntheses often show the African continent to be a particular blind spot in terms of data availability (e.g. Toming et al. (2020) for lake dissolved organic carbon, Johnson et al. (2022) for lake methane fluxes), and studies from Madagascar's inland waters are very scarce (Ralison et al., 2008; Marwick et al., 2014). Madagascar's lakes, are interesting as they lie within a region that combines tropical, subtropical, and anthropogenically influenced landscapes. These ecosystems are highly sensitive to environmental changes, such as deforestation, agricultural expansion, and climate variability, which affect carbon cycling processes and sediment fluxes. Understanding OC dynamics in Madagascar's lakes provides critical insights into carbon processing in subtropical regions with similar environmental pressures. This lack of data and uncertainty, especially in underrepresented regions like Madagascar, requires the collection of additional datasets over adequate spatial and temporal scales to refine global models of carbon cycling and better predict responses to anthropogenic and climatic changes.

Lake Alaotra is the largest freshwater system in Madagascar and is recognized as a hotspot of biodiversity. It is surrounded by marshes that provide the only remaining habitat of an endangered lemur species (*Hapalemur alaotrensis*), as well as by extensive floodplains that represent the most important rice-producing region of Madagascar (Lammers et al., 2015). Because of these high ecological, economical and scientific values, the Lake Alaotra wetland is recognized as a Ramsar site. The wetland marshes of Lake Alaotra occupy a surface area (~230 km$^2$) (Mietton et al., 2018) larger than the lake itself (currently ~200 km$^2$ of open water, Bakoariniaina et al., 2006; Ranarijaona, 2009), and are mainly located in the south-western part of the lake. Water and sediment transported through the rivers pass via floodplains and marshes before entering the lake. The characteristic hills in the Lake Alaotra watershed are currently dominated by grasslands but were likely to have been forested or characterized by wooded savannah vegetation up to ~2000 years ago (Broothaerts et al., 2022; Razanamahandry et al., 2022). A particular erosional feature called 'Lavaka' dots these landscapes - gullies which can reach very large dimensions (Brosens et al., 2022; Cox et al., 2010, 2023). Lavaka mainly occur in the central highlands of Madagascar, and their density is particularly high in the region of Lake Alaotra (Cox et al., 2010). Studies of $^{10}$Be in river sediments in central Madagascar indicated that sediments in the river are mainly lavaka-derived rather than colluvial sediments (Cox et al., 2009). In addition, since Lake Alaotra is located in one of the most important agricultural regions of Madagascar, it has been heavily impacted by human activities. The wetlands surrounding the lake have undergone significant alterations, primarily due to their conversion into rice paddies to support local agriculture. This process involves draining natural wetlands, altering their hydrology, and replacing native vegetation with cultivated rice crops. These changes have profoundly impacted the wetlands' ecological balance, biodiversity, and natural functions (such as water filtration, flood control, and providing critical habitats for biodiversity) (Lammers et al., 2015). This transformation has been accompanied by extensive deforestation along the hillslopes, which has increased sediment runoff. Furthermore, the construction of channels and dams for irrigation in the floodplains has disrupted natural hydrological processes, further modifying the state of the Lake Alaotra wetlands. Therefore, there appears an intuitive relationship between the forest disappearance, erosional processes, and the sedimentation of the lowlands (Kull, 2002). Erosion has a huge impact both in the upstream and downstream parts of a catchment. In the upstream regions, it does not only induce soil losses but in doing so, also degrades terrestrial ecosystems and their biodiversity (An et al., 2008; Montgomery, 2007; Zheng et al., 2005). Further downstream, the eroded soil is deposited in floodplains and lakes, and will affect the viability of aquatic ecosystems (Jenkins et al., 2010; Pattanayak and Wendland, 2007). There are indications that the productivity of rice in the Lake Alaotra basin dropped considerably to only about 40 % of its former level as a result of the silting of rivers and irrigation channels (Bakoariniaina et al., 2006). The reduction in rice production, along with demographic pressure, leads to increased rates of marshland conversion into agricultural fields or clearing for fishing. Therefore, the natural marshland vegetation has been reduced considerably, often by fire. Although some authors have claimed that eroded sediments have gradually filled Lake Alaotra and had reduced it to 60% of its original size by the 1960s (Bakoariniaina et al., 2006), recent data on the bathymetry and characteristics of the materials on the bottom of Lake Alaotra indicate that sedimentation in the lake itself remains non-significant (Ferry et al., 2013).

Studies on pollen from Lake Alaotra sediment archives have shown sedimentation rates of 0.3–0.6 mm y$^{-1}$ (Broothaerts et al., 2022), which are remarkably low given the high erosion rates in the catchment. This apparent discrepancy can be explained by the significant trapping of sediments in the floodplain and wetlands surrounding

the lake. According to Broothaerts et al. (2022), sedimentation rates in the floodplain are approximately 100 times
greater than in the lake, while wetlands exhibit rates about 10 times higher. These findings suggest that the
majority of sediments mobilized from hillslope erosion are deposited in upstream floodplain and wetland areas,
leaving relatively little to reach the lake itself.
Furthermore, sedimentation rates in the lake have remained consistently low over the last 1000 years, with no
significant increase observed despite extensive land-use changes in the catchment (Broothaerts et al., 2022). This
highlights the critical role of the floodplain and wetlands as sediment sinks, buffering the lake from excessive
sedimentation. This process will be further explored in the discussion and in a follow-up paper that examines
elemental and stable isotope data from sediment cores taken within the floodplain, marsh, and lake, which
reinforce these observations. In order to adequately interpret organic proxies within such sediment cores and link
them to changes in the catchment, a thorough understanding of potential carbon sources within the system is
critical, yet the contemporary functioning of Lake Alaotra has never been studied from a limnological or
biogeochemical perspective. This context provided an additional rationale for the current study, in which we adopt
a landscape-scale approach to investigate the different aquatic C pools and their stable isotope ratios, along with
a range of other physico-chemical and geochemical parameters, across the continuum of different inflowing water,
along the lake surface waters and in its outflow. Studies on pollen from Lake Alaotra sediment archives have
shown sedimentation rates of 0.3 - 0.6 mm y$^{-1}$ (Broothaerts et al., 2022), which is very low considering the high
erosion rates of the catchment. Moreover, during the last 1000 years, no significant increase in sedimentation rate
was observed (Broothaerts et al., 2022).

## 2 Materials and methods

### 2.1 Study Area

This study was conducted in the Lake Alaotra basin, Madagascar. Lake Alaotra is Madagascar's largest lake, and situated in the north-east of the island in the Toamasina province, between 17–18 °S and 48–49 °E and at an altitude of 775 m above sea level (Mietton et al., 2018) (Figure 1). The catchment of lake drains a catchment area of 4042 km$^2$ (Ferry et al., 2013); and the lake -and its wetlands and floodplains are surrounded by hills in an altitude range between 900 and 1300 m above sea level (Bakoariniaina et al., 2006). Grasslands form the dominant vegetation type in the Lake Alaotra catchment, and a high density of "lavaka" can be found across the landscape. On average, these reach dimensions of ~30 m wide, 60 m long and 15 m deep. The region experiences a tropical climate with two distinct seasons: a hot, rainy season from November to April and a cooler, dry season from May to October, which aligns with the division of our sampling period into (1) a dry season (June to October), characterized by cooler temperatures and lower precipitation, and (2) a rainy season (November to May), with higher temperatures and increased rainfall (Supplementary Figure S1). Annual rainfall ranges from 900 to 1250 mm, with the dry season contributing only 7–22% of the total. Rainfall typically peaks in January, with monthly totals exceeding 250 mm. The mean annual temperature in the Lake Alaotra region is 20.6°C, with average daily minima of 12°C in July and maxima of 28°C in January (Ferry et al., 2009).

Lake Alaotra is a shallow lake with an average water depth of 2–4m (Andrianandrasana et al., 2005). The open water surface of Lake Alaotra was less than 200 km$^2$ and freshwater water marshes cover around 230 km$^2$ (Bakoariniaina et al., 2006; Copsey et al., 2009), but these relative areas have varied over the years (Lammers et al., 2015). Lake Alaotra and its wetland marshes are surrounded by floodplains and ricefields covering around 820 km$^2$ (Ferry et al., 2009).

Lake Alaotra is filled by water mainly from infiltration, runoff, and flooding (Copsey et al., 2009). More than 20 rivers enter the lake, the largest of which are the Anony and Sahamaloto in the northwest, and the Sahabe and Ranofotsy in the southwest (Supplementary Figure S2). A network of man-made irrigation canals in the ricefields forms an extra connection between the rivers and the lake. The only outflow of the lake is the river Maningory, situated in the northeast of the lake (Figure 1 and Supplementary Figure S3).

In 1923, the construction of rice fields was initiated in the Lake Alaotra region. In the 1950s, dams and the delimitation of ricefields were constructed in order to improve the rice production capacity (Moreau, 1980). Most of the rivers flowing into the lake were progressively equipped with small hydraulic infrastructure to irrigate the ricefields towards the end of the 1980s. These generally consist of small water storage reservoirs and dams (e.g., Sahamaloto) with a large network of canals, thereby impacting the natural river network. Between 2003 and 2009, an additional reservoir was constructed (located in Andilanatoby on the river Ranofotsy) as part of an irrigation rehabilitation project. During the time of our fieldwork, a dam was constructed in the southeast on the river Sahabe.

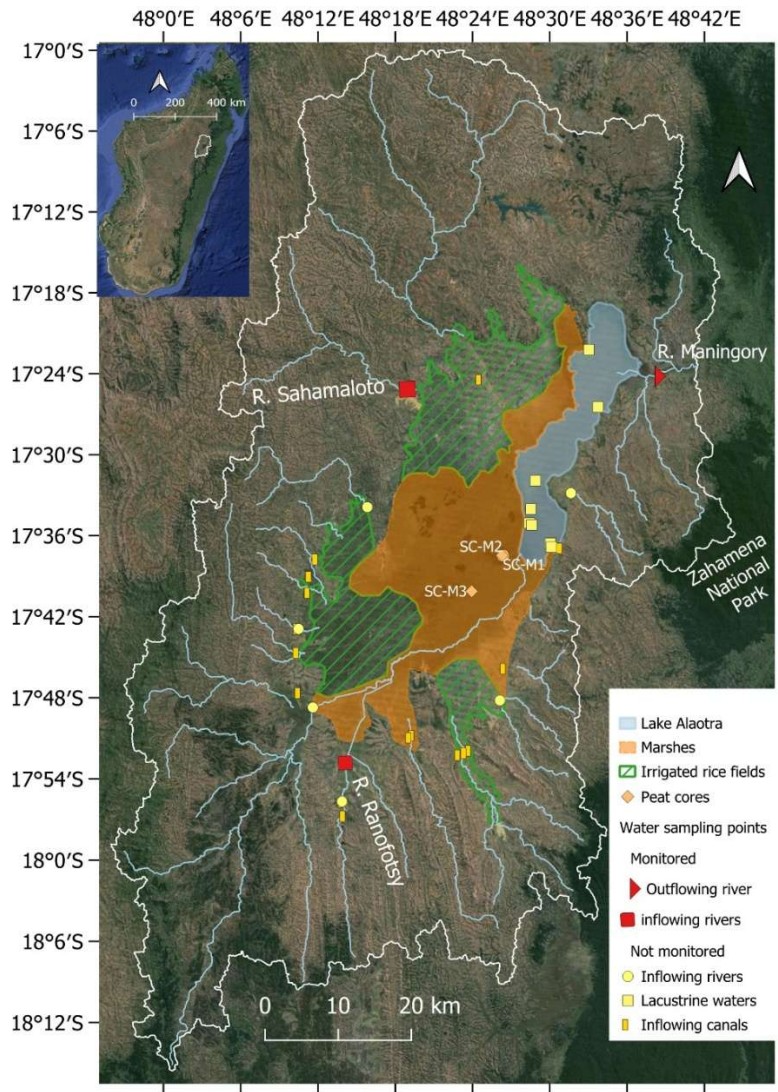

**Figure 1: Map of the Lake Alaotra catchment (delineated by the white line), indicating the location of sampling sites.**
**Lake Alaotra is indicated by blue filled polygon, wetlands are delineated by the orange polygon and the extent of**
**floodplains is indicated by the green dashed polygon. Background map taken from © Google Earth (2021).**
The monthly discharge of the Maningory was measured between the years of 1976 and 1986, ranging between 66
$m^3 s^{-1}$ and 315 $m^3 s^{-1}$ (Chaperon et al., 1993). Average discharge of water from the principal inflowing rivers basins
(basin of 4042 km²) to Lake Alaotra and the outflow Maningory have been calculated for the period between
1945–1979 (Chaperon et al., 1993; Dosseur and Ibiza, 1982). Results showed that there is a delay in the rise of
the annual peak discharge of approximately six weeks between the inflowing rivers and the Maningory. The
outflow presented a slower decrease in discharge compared to the rapid drop flows from March to April-May for
inflowing rivers, with a time difference of 6 weeks to 3.5 months (Figure 2). There was also a significant difference
between the runoff of the inflowing sub-catchments (500 mm) and the outlet catchment (340 mm).

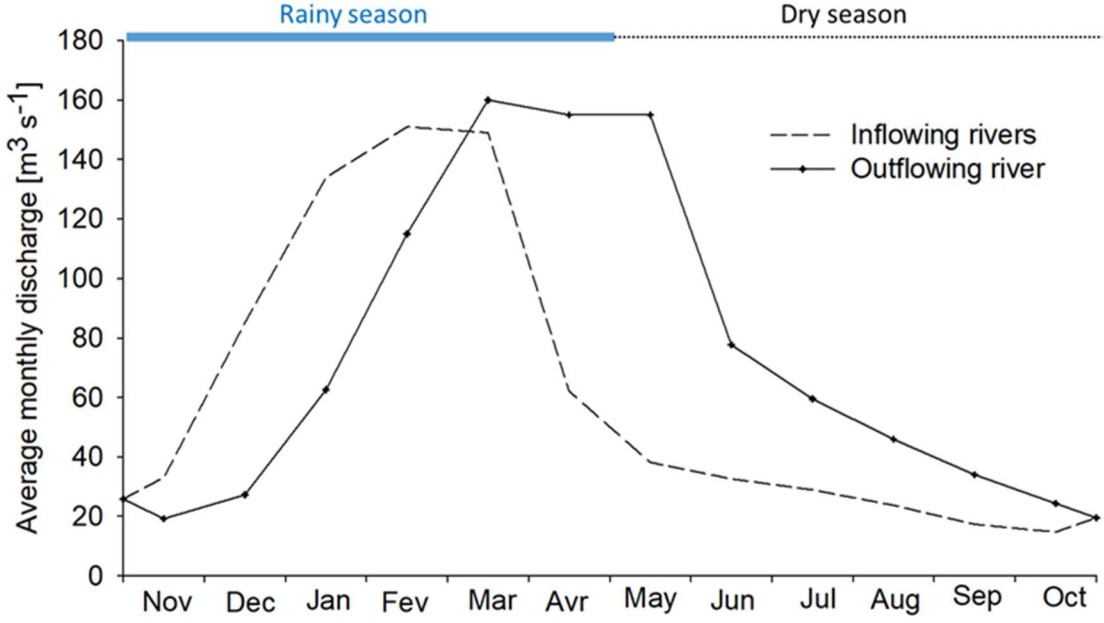


**Figure 2: Comparison between the average monthly discharge of inflowing rivers and outflowing river (Maningory) of**
**Lake Alaotra – data were collected between 1976 and 1987 (Chaperon et al., 1993). Continuous blue line and the dotted**
**black line represent the rainy season and dry season, respectively.**
The natural wetland and lake water body combined covers around 430 km², with wetlands largely located in the
southeast of the lake, while the inflowing rivers in the north (Anony) and those in the west are not surrounded by
substantial marshes before entering into the lake. These wetlands are seasonally flooded, and are dominated by
*Cyperus madagascariensis* or "zozoro" (Cyperaceae), covering ~50 % of the marshes (Lammers et al., 2015) (see
Supplementary Figure S4). This tall, robust, floating species requires either a permanent presence of a water
column (up to nearly 3 m deep) or at least a waterlogged environment. The population of *Cyperus*
*madagascariensis* in the Alaotra marshes has degraded due to the installation of rice fields and by its clearing for
traditional fishing (Ranarijaona, 2009).
The expansion of rice fields in the areas surrounding the lake has predominantly occurred through the conversion
of wetland ecosystems (Mietton et al., 2018). Since the era of cultivation, floodplains have been the main zone
where farmers in Lake Alaotra cultivate rice (especially irrigated rice) (Supplementary Figure S5). During the
rainy season, floods regularly occur in the Alaotra plain and lead to strong siltation over the ricefields (Ferry et
al., 2013).
Around 750 000 people live in the area of Lake Alaotra (estimate for 2011), for whom rice cultivation and fishing
are important sources of livelihood (Penot et al., 2012), a stark increase from ~110,000 people in the 1960s
(CREAM, 2012). Due to this demographic pressure (Jacoby and Minten, 2007), agricultural land is becoming
limited, forcing many people to convert the marshes to ricefields (Lammers et al., 2017). This practice of
cultivation consists of growing rice in shallow lake water and by converting the marshes at the lake edge. In
addition, farmers have started to use the hillslopes for the production of upland rice, maize, peanuts and cassava
and a range of vegetables (Penot et al., 2018).

## 2.2 Sampling design

### 2.2.1 River water sampling and monitoring

Water samples from inflowing rivers and canals were collected during our first and second fieldwork campaigns: (i) April–June 2018 (dry season) and (ii) January–March 2019 (rainy season). Due to low water levels during the dry season, sampling was limited to six rivers with sufficient flow. In contrast, the rainy season campaign benefited from higher flow conditions, allowing for weekly sampling across most rivers and canals. Sites were chosen based on accessibility and their hydrological significance in representing catchment-wide inflow to the lake.

Biweekly monitoring of select rivers and the lake outlet was conducted from April 2018 to August 2019 to assess seasonal changes in water parameters. Rivers selected for monitoring—Sahamaloto and Ranofotsy—were chosen because they drain grassland-dominated catchments and are accessible before entering the floodplain. The Maningory River, the only outlet of Lake Alaotra, was monitored at the bridge approximately 5 km downstream of the lake outlet, which is assumed to largely reflect the lake outflow characteristics, although periodic inputs by the local riparian environment cannot be excluded.

### 2.2.2 Lake water sampling

Lake sampling was carried out along a North-South gradient during field campaigns, covering both nearshore and open-water areas. Water samples were collected at a depth of ~0.5 m using a Niskin bottle to ensure consistency across sites. This sampling depth was chosen to represent surface water conditions where primary productivity and particulate organic carbon (POC) are most active.

### 2.2.3 Marsh vegetation sampling

Marsh sampling was included to examine the role of wetlands as contributors of organic matter to the lake. Marsh plant samples were prioritized over terrestrial plants because they were the dominant vegetation in the wetland zones adjacent to the lake. Terrestrial plant data were referenced from existing studies (e.g., Razanamahandry et al., 2022). Phytoplankton were not sampled directly due to methodological constraints but constraints on their expected stable isotope composition will be discussed based on $\delta^{13}C$ measurements of DIC and POC, and their potential importance is represented indirectly through *in situ* measurements of the primary production in the lake water.

### 2.2.4 Sediment cores

Marsh sediment cores were collected from marshes to study historical deposition patterns and organic matter sources over time. This core provides a time-resolved view of biogeochemical processes that cannot be captured with surface samples. For the scope of this paper, data from these depth profiles will be summarized to present an end-member value for marshland-derived OC inputs into the lake.

**2.2.5 Rainfall data**
Rainfall data were obtained from meteoblue.com (dataset spanning 40 years), a meteorological service that
employs weather models based on the NMM (Nonhydrostatic Meso-Scale Modeling) technology. We selected
two locations: Tanambe (northwest of Lake Alaotra) and Ambatondrazaka (southeast of Lake Alaotra).
**2.3. Field and laboratory analyses**
Water temperature, conductivity, dissolved oxygen, and pH were measured *in situ* using a Yellow Springs
Instruments (YSI) ProPlus probe in surface waters ($\leq$0.5 m).
**2.3.1 $CH_4$ measurement and analysis**
Samples for the determination of $CH_4$ and were collected with a Niskin bottle in surface waters ($\leq$0.5 m) and the
water was transferred, taking care to avoid the inclusion of air bubbles, with a silicone flexible tube into two 60
ml borosilicate serum bottles (Wheaton$^{TM}$), preserved with 200 µl of a saturated solution of $HgCl_2$, sealed with a
butyl stopper without a headspace and crimped with an aluminium cap, for subsequent analysis at the laboratory.
Measurements of $CH_4$ dissolved concentrations were made with the headspace technique (Weiss, 1981) with a
gas chromatograph (GC) on a headspace (20 ml) made with ultra-pure $N_2$ (Air Liquide$^{TM}$ Belgium) in two 60 ml
serum bottles for a replicate analysis of $CH_4$. The GC (SRI$^{TM}$ 8610C) was equipped with a flame ionisation detector
for $CH_4$, and was calibrated with $CH_4$:$N_2$ gas mixtures with mixing ratios of 1, 10 and 30 ppm manufactured and
certified at ±2% by Air Liquide$^{TM}$ Belgium. The precision based on duplicate samples was better than ±4% and
the detection limit was 0.5 nmol L$^{-1}$.
**2.3.2 $pCO_2$ analysis**
To measure $pCO_2$, 60 mL syringes were filled either directly with surface water ($\leq$0.5 m) from the river and lake
or from the Niskin bottle. An additional syringe was filled with ambient air. A 30 mL headspace (ambient air)
was created, and after 10 minutes of vigorous shaking, the headspace was injected into a LICOR LI-820 infrared
gas analyser (Borges et al., 2015). Calibration of the LICOR was performed before and after each sampling
campaign with ultrapure $N_2$ and a standard (Air Liquide Belgium) with a $CO_2$ mixing ratio of 1019 ppm (Air
Liquide$^{TM}$ Belgium). The precision of $pCO_2$ measurements was estimated to be ± 5 %, and the detection limit was
3 ppm. For logistical reasons, $pCO_2$ measurements were conducted in situ during field campaigns only, using the
setup described above. However, regular monitoring was limited to a subset of rivers and was performed by a
separate team without access to the same equipment, resulting in gaps in the dataset. Additionally, not all rivers
sampled during the field campaigns were included in the regular monitoring program, further contributing to the
observed gaps.
**2.3.3 TSM, POC, PN and $\delta^{13}$C-POC**
Samples for TSM (total suspended matter), POC (particulate organic C), PN (particulate nitrogen) and stable
isotope ratios in POC ($\delta^{13}$C-POC) involved collection of water samples in the centre of the rivers by using a Niskin
bottle. Samples for TSM were obtained by filtering a known volume of water (approximately 100 to 250 mL of

water) on pre-weighed and pre-combusted (450°C) 47 mm Whatman GF/F filters with a nominal pore size of 0.7 μm and then air-dried. These were later oven-dried prior to weighing to calculate TSM loads.

A known volume (20–50 mL) of water was filtered through pre-combusted (450°C) 25 mm Whatman GF/F filters to determine the concentrations of POC and PN, and $\delta^{13}$C-POC. These filters were air-dried after collection, and later treated with concentrated HCl fumes in a desiccator for four hours to eliminate inorganic C. Afterwards, the filters were dried in the oven at 50°C and packed in Ag cups. The analysis was conducted using an Elemental Analyser Isotope Ratio Mass Spectrometer (EA-IRMS: Thermo Flash HT/EA and Delta V Advantage) setup. Calibrations of concentrations and $\delta^{13}$C data were based on certified caffeine (IAEA-600) and two in-house references: leucine and tuna muscle tissue (previously calibrated versus certified standards). Reproducibility of $\delta^{13}$C measurement was better than ± 0.2 ‰. POC/PN ratios are reported as mass/mass ratios.

$\delta^{15}$N data were measured alongside POC, PN, and $\delta^{13}$C, and while they are not discussed in detail in this study, the full data are provided in the supplementary material to ensure the data could be used in future research.

**2.3.4 DOC and $\delta^{13}$C-DOC**

To determine the dissolved organic C (DOC) concentration and $\delta^{13}$C-DOC values, 40 mL of filtered water samples (first filtered with pre-combusted (450°C) 47 mm Whatman GF/F filters with a pore size of 0.7 μm and subsequently with 0.2 μm syringes filters) were collected and stored in glass vials with Teflon-coated screw caps. To preserve the water samples, 100 μL of $H_3PO_4$ was added. Analysis of DOC and $\delta^{13}$C-DOC was performed on a wet oxidation TOC analyzer (IO Analytical Aurora 1030W) coupled with an isotope ratio mass spectrometer (Thermo Finnigan Delta XP). Quantification and calibration were performed with IAEA-C6 ($\delta^{13}$C=-10.4 ‰) and an internal sucrose standard ($\delta^{13}$C=-26.99 ± 0.04 ‰).

**2.3.5 Total alkalinity (TA)**

Total alkalinity (TA) was measured via an open-cell titration with 0.1 mol $L^{-1}$ HCl (Gran, 1952) on 50 mL water samples filtered on 0.2 μm. Data quality was verified based on certified reference material from Andrew Dickson (Scripps Institution of Oceanography, University of California, San Diego, USA). Typical reproducibility of TA measurements was better than ± 3 μmol $L^{-1}$.

**2.3.6 DIC and $\delta^{13}$C-DIC**

To measure $\delta^{13}$C-DIC, water was transferred directly from the Niskin sampler and stored air-free in 12 mL glass vials. Samples were poisoned with 20 μL of a saturated $HgCl_2$ solution. Analysis of $\delta^{13}$C-DIC was done one day after a He (helium) headspace of 2.5 mL was created. To convert DIC to $CO_2$, 100 μL of acid $H_3PO_4$ (99 %) was added into the vials, followed by an overnight equilibration. Approximately 1 mL of the headspace was then injected into the He flow of the EA-IRMS setup described above. $\delta^{13}$C values were corrected for isotope fractionation between the dissolved $CO_2$ in the water and the $CO_2$ in the created headspace and for the partitioning of $CO_2$ between the two phases as described in Gillikin and Bouillon (2007).

The DIC concentration was calculated with the Excel Macro CO2SYS (V2.1) created by Lewis et Wallace (1998) in which values of water temperature, TA measurement, and pCO2 direct measurement are the inputs.

### 2.3.7 Chlorophyll a

The concentration of Chlorophyll a was determined from the extraction of pigments from filtered lake water. A known volume of water was filtered through pre-combusted (450°C) 47 mm Whatman GF/F filters of 0.7 μm and later stored in a freezer until analysis. High performance liquid chromatography (HPLC) was used to determine pigment concentrations. Pigments were extracted in 10 mL of 90 % HPCL grade acetone. The pigment extract was stored in 2 mL amber vials at -25°C prior to a two sonification steps of 15 minutes separated by an overnight period at 4°C. The gradient elution method described by Wright et al. (1991), combined with a Waters system comprising a photodiode array and fluorescence detectors were used to perform the HPLC analyses. Calibration is based on commercial external standards (DHI Lab Products, Denmark). Typical reproducibility of pigment concentration measurement was better than 7 %. The CHEMTEX software (CSRIO Marine Laboratories) based on input ratio matrices adapted for freshwater phytoplankton is used to process pigment concentration data.

### 2.3.8 Primary production rates

We measured the pelagic primary production (PP) rate in the lake by *in situ* $^{13}C$ incubations at different light intensities. First, a solution of 500mL of surface water spiked with Na $H^{13}CO_3$ was prepared. A subsample of this solution was transferred and preserved in triplicate 12 mL exetainer vials and immediately poisoned with saturated 20 μL solution of $HgCl_2$ to measure the initial $\delta^{13}$C-DIC value of the spiked water. Eight 50 mL polycarbonate flasks were filled with the spiked solution and were organized into a floating incubator with different filters to provide light shading from 0 to 90 % natural mid-day light energy. An Odyssey photosynthetic irradiance recording system (Photosynthetic active radiance (PAR) logger) was used to monitor the incident light during the entire period of the field campaign. At the end of the incubation, which lasted at least two hours, we added 100 μL of formalin to instantly stop the biological activity. One supplementary bottle was processed in a similar way at the beginning and at the end of the incubation to produce a dark incorporation control. Each water sample was then filtered on a pre-combusted (450°C) 25mm Whatman GF/F filter to collect the particulate fraction. These PP incubations were performed 4 times during the first campaign (April–July 2018), 2 times during the second fieldwork campaign (January 2019–March 2019) and 3 times during the third campaign (August–October 2019). In addition, PP incubations were performed in the reservoirs of Andilanatoby (in the south of the catchment) and Sahamaloto (in the northwest of the catchment) during the first and second campaign (Figure 1). Analyses of $\delta^{13}$C-POC and $\delta^{13}$C-DIC for primary production samples followed the same procedures as described earlier, but given the high $^{13}$C-enrichments in the DIC pool, the obtained $\delta^{13}$C values were not corrected for isotope fractionation between gaseous and dissolved $CO_2$.

To calculate the specific photosynthetic rate in each individual bottle *i*, $P_i$ (in μg C $L^{-1}$ $h^{-1}$), we followed Dauchez et al. (1995) based on the initial and final $\delta^{13}$C-POC values and $\delta^{13}$C-DIC of the spiked incubated solution, and assuming that isotopic discrimination is negligible (Legendre and Gosselin, 1997).

For each experiment, the maximum specific photosynthetic rate $P_m$ (in μg C $L^{-1}$ $h^{-1}$) and the irradiance at the onset of light saturation $I_k$ (μE $m^{-2}$ $s^{-1}$) were obtained by fitting $P_i$ into the irradiance gradient provided by the incubator $I_i$ (μE $m^{-2}$ $s^{-1}$), using the following Vollenweider's equation, with a=1 and n=1, allowing for photoinhibition (Vollenweider, 1966):

$$P_i = 2P_m \left[ \frac{I_i/2I_k}{1+(I_i/2I_k)^2} \right], \tag{Eq.1}$$

Where $P_i$ is the photosynthetic rate in bottle $i$ during the incubation time and $I_i$ is the corresponding mean light
during the incubation. Fitting was performed using the Gauss-Newton logarithm for nonlinear least squares
regression. Daily depth-integrated primary production (mg C m$^{-2}$ day$^{-1}$) was calculated according to Kirk (1994)
using the following equation:
$P(z, t) = 2P_m[\frac{I(z,t)\,/2I_k}{1+(\,I(z,t)\,/2I_k)^2}],$                                        (Eq.2)
Where P(z, t) is the photosynthesis at depth z and time t, and I (z, t) is the underwater light determined from Ke
and surface irradiance recorded every 5 min and assuming a vertically homogenous Chl-a profile. Assuming that
short-term incubation provides an estimate which is close to gross primary production (GPP), we calculated water
column daily respiration (R, mg C m$^{-2}$ day$^{-1}$) as in Reynolds (2006), considering a respiration rate of 0.16 mg C
mg Chl-a h$^{-1}$ at 18°C (based on López-Sandoval et al. (2014) a $Q_{10}$ of 2 for adjusting for lake temperature, a
constant respiration rate over 24 hours, and the whole lake depth at the study sites).
**2.4 OC analysis of vegetation and marsh sediment cores**
Different species of common marsh plant species were sampled. Vegetation samples were air-dried in the field,
then dried in oven at 50°C in the laboratory. A mortar, pestle and nitrogen liquid were used to grind dried
vegetation samples into a well homogenised powder.
Sediment cores in Lake Alaotra marsh (Figure 1) were collected with an UWITEC gravity corer adapted for
manual coring with 2 m sampling tubes (Ø = 6cm). SC-M1 and SC-M2 were collected in the eastern part of the
marshes which are permanently waterlogged, while core SC-M3 was collected further south within the marshes
at a location which is not waterlogged throughout the year (Figure 1). All cores were sliced at a resolution of 1
cm. Samples were stored in a portable freezer at -18°C for preservation. Afterwards, sediment core samples were
freeze-dried and homogenised in order to take subsamples for laboratory analysis.
Subsamples were weighed into Ag cups to determine OC content, total nitrogen content, and $\delta^{13}$C and $\delta^{15}$N of
organic matter. All subsamples (except for vegetation) were acidified with 40 µL HCl (10 %) to eliminate all
inorganic C. OC content, total nitrogen content and $\delta^{13}$C of OC were measured as described above for POC, PN
and $\delta^{13}$C-POC.
**3 Results**
**3.1 Seasonal water level variability**
The discharge of inflowing rivers and canals (inflowing water) respond strongly to the seasonality in precipitation
in the catchment, thus water levels were high mainly during the rainy season and low (up to dry conditions) during
the dry season (Figure 3). Water levels of the Maningory (lake outlet) also varied seasonally (Figure 3) but with
a much smoother cycle, being lowest in December and highest in May of the year of sampling.

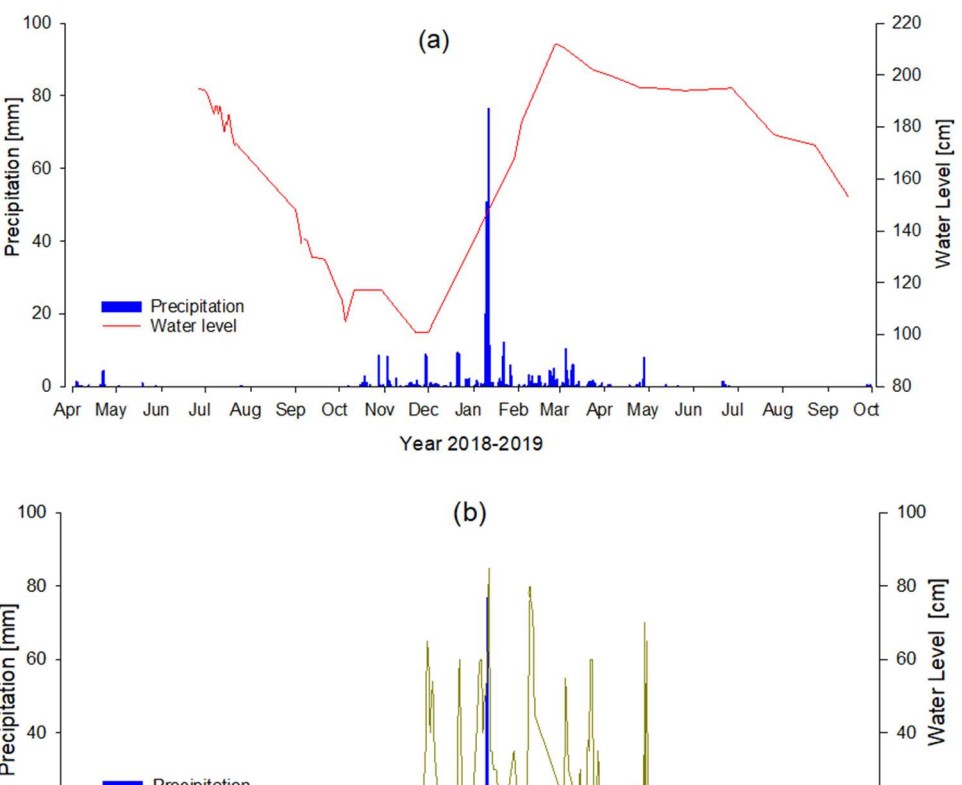

**Figure 3: Variation of water levels (full line, right Y-axis; our data) of the Maningory (top panel) and the Ranofotsy (lower panel), plotted along with the amount of daily precipitation (bars, left Y-axis) in the region during sampling period. Precipitation data were obtained from meteoblue (www.meteoblue.com).**

**3.2 Dissolved oxygen**

The saturation level of dissolved oxygen (expressed in %) in inflowing water showed a higher value during the dry season (87.3 ± 18.2%) compared to the rainy season (77.0 ± 14.5%). No seasonal variation was observed for the saturation of dissolved oxygen in lacustrine water, and the values ranged between 58.4 and 104.5% with an average value of 83.4%. The saturation of dissolved oxygen in the outlet waters was higher during the dry season, with an average value of 71.2 ± 29.3% compared to the value during the rainy season (47.1 ± 28.5%). The saturation levels of dissolved oxygen inflowing and lacustrine water were higher compared to the saturation level of outflowing water.

**3.3 pH and dissolved inorganic carbon (DIC)**

Inflowing water showed pH values of 7.2 ± 0.5, increasing slightly in Lake Alaotra (7.4 ± 0.6), but lower values were found in the Maningory (outlet; 6.9 ± 0.6). The DIC concentrations for inflowing water varied between 213 and 2149 $\mu mol \ L^{-1}$, with an average value of 690 ± 158 $\mu mol \ L^{-1}$, and no significant seasonal variation was observed. The trends in DIC concentrations of lacustrine water showed a seasonal variation with a higher value during the rainy season (812 ± 106 $\mu mol \ L^{-1}$) compared to the dry season (608 ± 134 $\mu mol \ L^{-1}$). For the water in the lake outlet, the DIC concentration values ranged between 485 and 931 $\mu mol \ L^{-1}$, with an average of 768 $\mu mol$

L⁻¹, and no seasonal variation was observed. There was a significant difference between the DIC concentrations of the inflowing and outlet waters. The DIC concentration at the outlet showed a higher value.

### 3.4 pCO₂ and CH₄ dynamics

The $pCO_2$ values were higher in the lake outflow (7242 ± 3709 ppm) than in the inflowing rivers and canals (3507 ± 2861 ppm). The lowest $pCO_2$ values were observed in the lake (1595 ± 881 ppm) (Figure 4). Dissolved $CH_4$ concentrations were greater in the lake outflow (2040 ± 804 nmol L⁻¹) than in the inflowing rivers and canals (1157 ± 1986 nmol L⁻¹). The lowest value of dissolved $CH_4$ were found in the lake (91 ± 171 nmol L⁻¹). $pCO_2$ and $CH_4$ were correlated across the whole datasets (Figure 5). $pCO_2$ and dissolved $CH_4$ concentrations increased during the rainy season as shown in Figure 6.

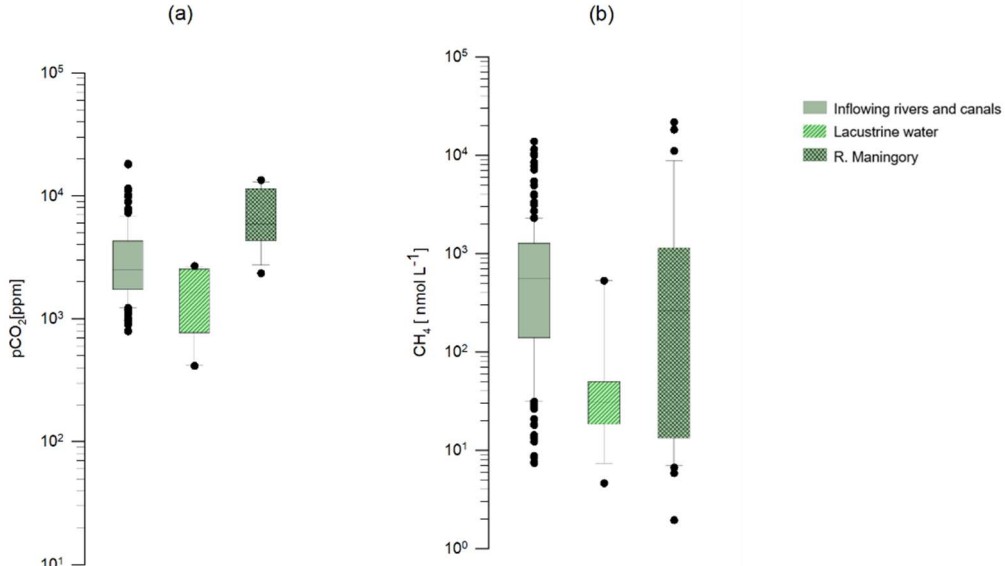

**Figure 4: Boxplot of partial pressure of CO₂ (pCO₂) (a) and dissolved CH₄ concentration (b) of inflowing rivers and canals, Lake Alaotra, and the Maningory River (lake outflow).**

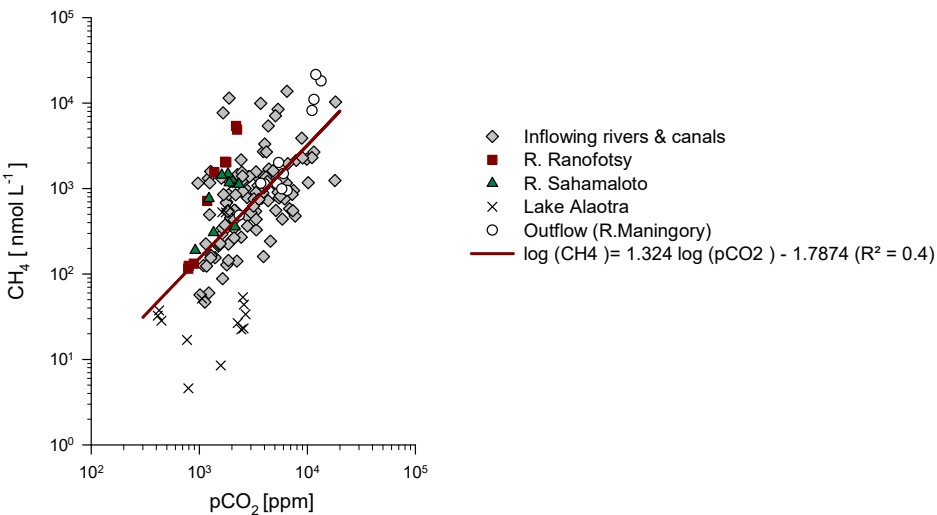

**Figure 5: Relation of CH₄ versus pCO₂ of water samples. Continuous red line represents the regression line of CH₄ versus pCO₂ in log-log scale.**

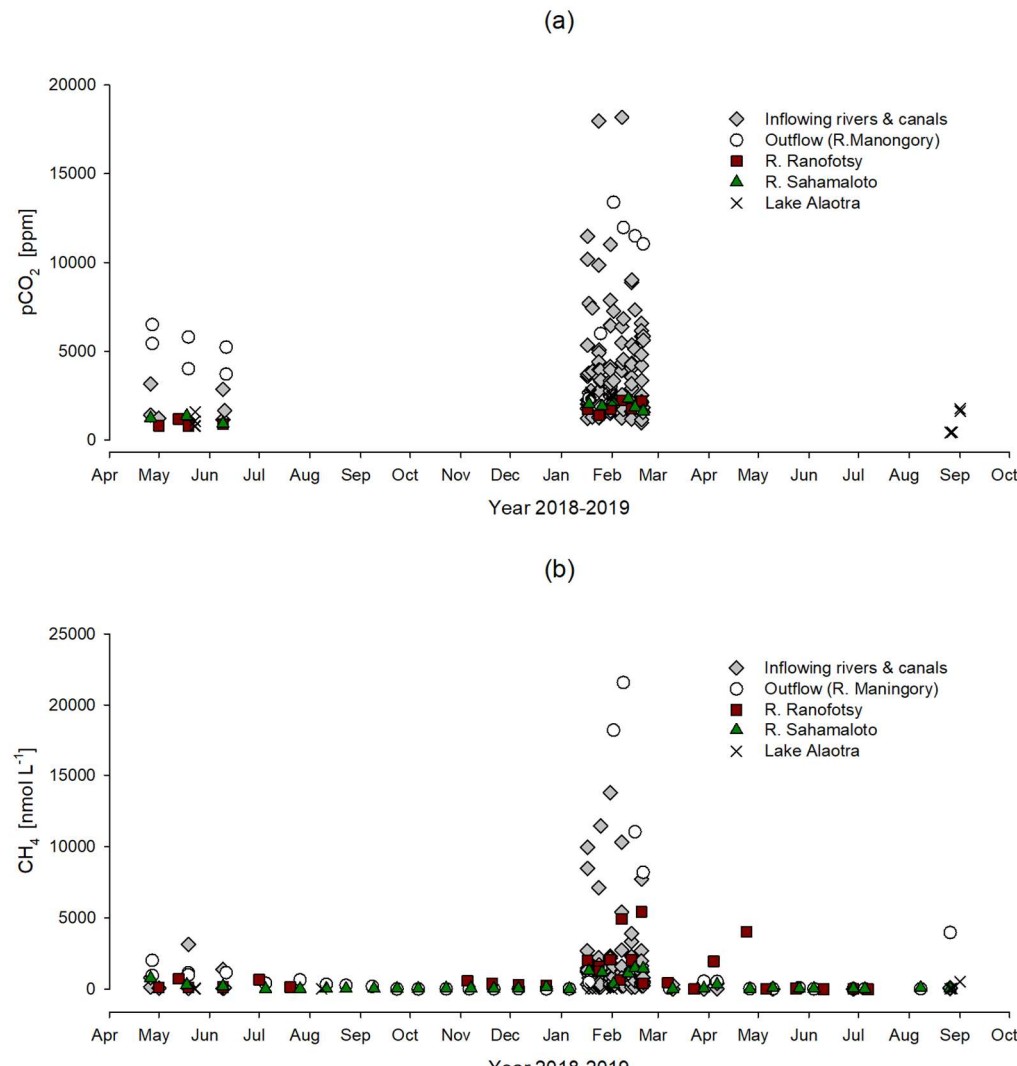

420

**Figure 6: Seasonal variation of (a) pCO₂ (expressed in ppm) and (b) CH₄ (expressed in nmol L⁻¹) water from inflowing rivers, canals and lake, lake outflow (Maningory) in the Alaotra Lake system (Madagascar) between May 2018 and September 2019.**

## 3.5 Total suspended matter (TSM)

TSM concentrations of the inflowing water ranged between 3.7 and 215 mg L$^{-1}$ during the dry season and between 3.7 and 1392.5 mg L$^{-1}$ during the rainy season (Table 1), reaching maximum values during the middle of the rainy season (January and February). Within Lake Alaotra, TSM varied between 5.8 and 46.7 mg L$^{-1}$ during the dry season and between 8.5 and 39.3 mg L$^{-1}$ during the rainy season. At the outlet (Maningory), TSM ranged between 11 and 60 mg L$^{-1}$ and between 5.8 and 115.2 mg L$^{-1}$ during the dry and rainy seasons, respectively. The highest TSM in the Maningory were reached at the start of the rainy season when the water levels were the lowest.

## 3.6 Particulate organic carbon (POC) and its characteristics

POC concentrations of inflowing water varied between 0.3 to 2.4 mg C L$^{-1}$ and between 0.5 to 7.5 mg C L$^{-1}$ during the dry season and rainy seasons, respectively, reaching maximal values in the middle of the rainy season (Figure 7a). POC concentrations in Lake Alaotra ranged between 1.4 and 6.5 mg L$^{-1}$, with an average value of 3.0 ± 1.7

mg L$^{-1}$, and increased further at the Maningory outlet (1.2 to 10 mg L$^{-1}$) with an average value of 4.2 ± 2.4 mg L$^{-1}$.

The contribution of POC to the TSM loads of inflowing water (%POC) was on average 8.3 ± 8.8 % and ranged between 0.1 and 34.8 % during the dry season. During the rainy season, %POC varied between 0.4 to 26.5 % with an average of 2.4 ± 4.0 %. The contribution of POC to the TSM was much higher within Lake Alaotra, ranging between 12.0 and 28.9 % with no clear seasonal variation (20.8 ± 6.4 % and 21.4 ± 6.9 % during the dry and rainy season, respectively). In the lake outlet, %POC varied in a narrow range and was significantly higher compared to that of rivers: on average 14.4 ± 8.5 % and 10.7 ± 7.2 % during the dry and rainy seasons, respectively.

$\delta^{13}$C-POC of inflowing waters ranged between -28.3 and -19.0 ‰ (Figure 8) with an average of -24.6 ± 1.4 ‰. The average $\delta^{13}$C-POC value of inflowing water during the rainy season (-24.3 ± 1.9 ‰) was slightly higher than that during the dry season (-25.6 ± 1.3 ‰), whereas no seasonality was evident in Lake Alaotra (-26.5 ± 2.1 ‰). $\delta^{13}$C-POC at the outlet varied between -28.3 and -21.9 ‰ with an average of -25.3 ± 1.4 ‰. The suspended organic matter pool of inflowing waters was characterized by an average POC/PN of 10.3 ± 2.6 while the POC/PN ratio of the outlet averaged 9.5 and varied between 6 and 12.2. The POC/PN ratio of lacustrine water ranged from 8.3 to 11.3.

Table 1: TSM and %POC (minimum, maximum and mean values) of samples from inflowing water, Lake Alaotra, and for the Maningory (lake outlet) during rainy and dry season.

|  | Season | Group | Min | Max | Mean ± std dev (n) |
|---|---|---|---|---|---|
| **TSM** | Dry | inflowing water | 3.7 | 215.8 | 44.0 ± 52.2 (n=42) |
|  |  | Lake waters | 5.8 | 46.7 | 18.8 ± 15.0 (n=10) |
|  |  | River Maningory | 11.0 | 60.0 | 31.7 ± 16.5 (n=20) |
|  | Rainy | inflowing water | 3.7 | 1392.5 | 131.9 ± 224.2 (n=74) |
|  |  | Lake waters | 8.5 | 39.3 | 16.6 ± 11.8 (n=6) |
|  |  | River Maningory | 5.8 | 115.2 | 47.5 ± 37.9 (n=18) |
| **%POC** | Dry | inflowing water | 0.1 | 34.8 | 8.3 ± 8.8 (n=40) |
|  |  | Lake waters | 12.0 | 28.8 | 20.8 ± 6.4 (n=8) |
|  |  | River Maningory | 2.9 | 36.2 | 14.4 ± 8.5 (n=19) |
|  | Rainy | inflowing water | 0.4 | 26.8 | 3.5 ± 4.0 (n=69) |
|  |  | Lake waters | 10.1 | 28.9 | 21.4 ± 6.9 (n=6) |
|  |  | River Maningory | 2.9 | 31.3 | 10.7 ± 7.2 (n=17) |

### 3.7 Dissolved organic carbon (DOC) and $\delta^{13}$C values

DOC concentrations of the inflowing waters ranged between 1.0 and 8.1 mg C L$^{-1}$ (Figure 7b, Figure 8) with an average value of 2.6 ± 1.1 mg C L$^{-1}$. DOC concentrations of the two monitored rivers (Ranofotsy and Sahamaloto) ranged between 1.0 and 5.3 mg C L$^{-1}$, and no significant seasonal variation was observed (Figure 7). DOC concentrations of lacustrine water varied between 7.5 and 18.6 mg L$^{-1}$ where the maximum DOC concentrations were obtained during the rainy season (Figure 7b). The DOC concentration of the Maningory river varied between 7.6 and 13.7 mg L$^{-1}$ with an average of 9.5 ± 1.4 mg L$^{-1}$.

The $\delta^{13}$C-DOC of inflowing water (river and canals) values ranged between -30.9 and -15.2 ‰ with an average value of -23.0 ± 2.1 ‰. The $\delta^{13}$C-DOC of lacustrine water ranged between -24.0 and -20.3 ‰ with an average

value of -22.2 ± 1.1 ‰. The δ¹³C-DOC at the outlet varies between -23.6 and -19.4 ‰ with an average value of -

21.4 ± 0.1 ‰ (Figure 7 and Figure 8).

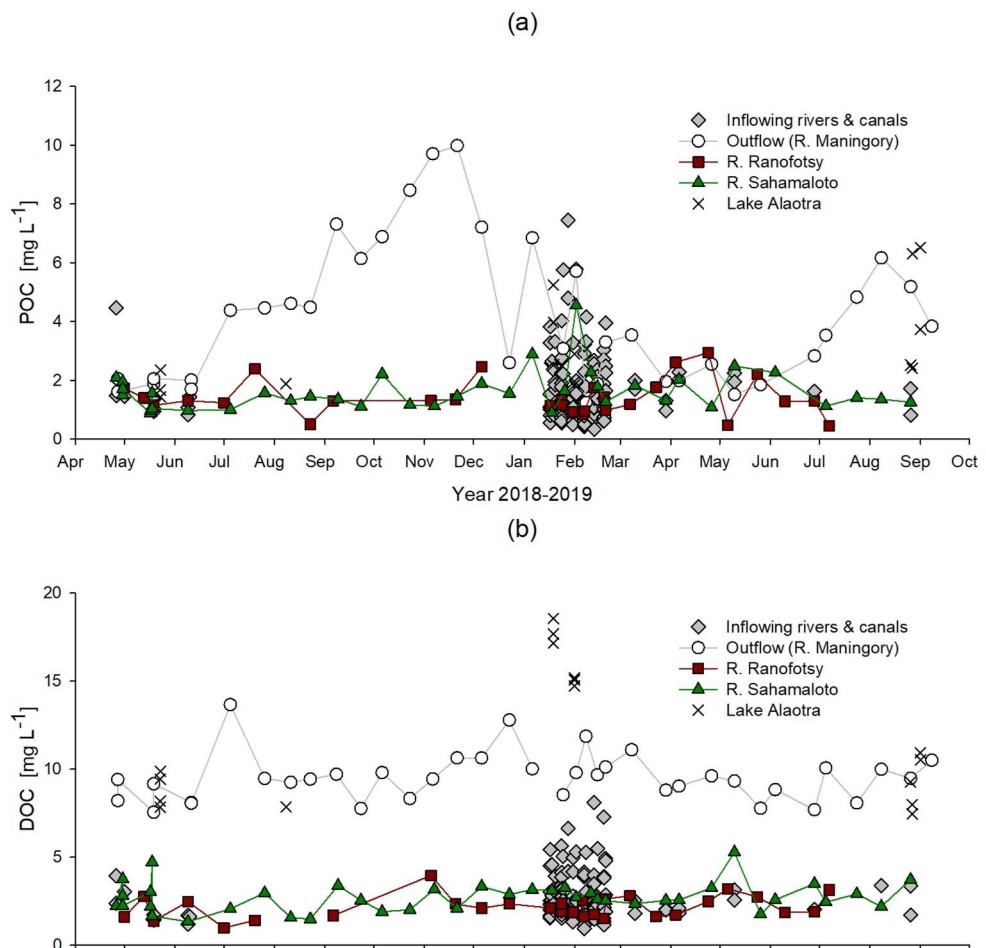

**Figure 7: Seasonal variation of (a) particulate organic carbon (POC) and (b) dissolved organic carbon (DOC)**
**concentrations (expressed in mg C L⁻¹) of water from inflowing rivers, canals and lake, lake outflow (Maningory) in**
**the Alaotra Lake system (Madagascar) between May 2018 and September 2019.**

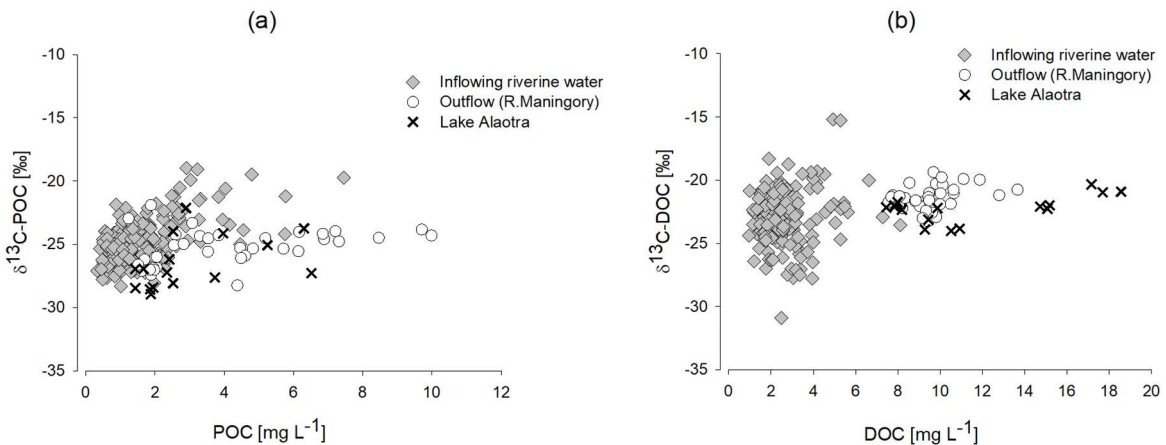

**Figure 8: Stable isotope composition versus concentrations for (a) particulate organic carbon POC and (b) dissolved**
**organic carbon DOC for inflowing water, Lake Alaotra, and the lake outflow (Maningory).**

### 3.8 Chl-a concentrations

Mean Chl-a concentration of the lake range between 8.0 and 10.0 µg L$^{-1}$ (Table 2) and the mean daily net primary production (NPP) was estimated at 538.3 mg C m$^{-2}$ day$^{-1}$ (range, 144.5 and 1250 mg C m$^{-2}$ day$^{-1}$). For TSM, POC, $\delta^{13}$C-POC, DOC, $\delta^{13}$C-DOC and POC/PN values, paired t-tests did not reveal any significant differences between Lake Alaotra and the lake outflow (Maningory).

Table 2: Chlorophyll a (Chl-a), POC, and POC/Chl-a ratios of lacustrine water (Lake Alaotra).

| Date | Chl-a (µg L$^{-1}$) | POC (mg L$^{-1}$) | POC/Chl-a (mg:mg) |
|---|---|---|---|
| May 2018 | 8.5 | 1.7 | 196 |
| May 2018 | 10.0 | 1.4 | 143 |
| January 2019 | 9.3 | 5.2 | 564 |
| January 2019 | 8.0 | 4.0 | 498 |

### 3.9 OC content and δ13C-OC, OC/TN in marsh sediment cores

The marsh sediment cores exhibit a range of organic carbon content (%OC), stable isotope values ($\delta^{13}$C-OC), and OC/TN ratios (summarized in Table 3). SC-M1 shows consistent %OC values, with a mean of 22.2% (range: 11.5%–36.2%) and $\delta^{13}$C-OC values averaging -19.6 ‰ (range: -22.4‰ to -17.0‰). The OC/TN ratios for SC-M1 are relatively stable, with a mean of 16.74 (range: 13.4–21.6). SC-M2 presents a similar pattern, with %OC averaging 20.6% (range: 10.6%–37.0%), $\delta^{13}$C-OC values of -19.5‰ (range: -20.0‰ to -19.0‰), and OC/TN ratios averaging 17.0 (range: 15.1–19.6). SC-M3, however, demonstrates greater variability, with %OC ranging widely from 5.7% to 58.8% (mean: 29.9%), $\delta^{13}$C-OC values averaging -17.9‰ (range: -23.1‰ to -15.6‰), and OC/TN ratios showing a broad range of 14.0 to 47.1 (mean: 23.6).

Table 3: Summary of OC content, $\delta^{13}$C-OC, and OC/TN ratios in marsh sediment cores

| Core | % OC | | | $\delta^{13}$C-OC | | | OC/TN | | |
|---|---|---|---|---|---|---|---|---|---|
| | SC-M1 | SC-M2 | SC-M3 | SC-M1 | SC-M2 | SC-M3 | SC-M1 | SC-M2 | SC-M3 |
| **Mean** | 22.2 | 20.6 | 30.0 | -19.6 | -19.5 | -17.9 | 16.7 | 17.1 | 23.6 |
| **S.D.** | 7.3 | 6.9 | 18.7 | 1.2 | 0.3 | 1.4 | 2.2 | 1.0 | 9.0 |
| **Median** | 21.5 | 18.3 | 34.4 | -19.7 | -19.6 | -18.0 | 16.7 | 17.0 | 21.2 |
| **Min.** | 11.5 | 10.6 | 5.7 | -22.4 | -20.1 | -23.1 | 13.4 | 15.1 | 14.0 |
| **Max.** | 36.2 | 37.0 | 58.8 | -17.0 | -19.0 | -15.6 | 21.6 | 19.6 | 47.1 |

### 3.10 OC content and $\delta^{13}$C-OC of marsh vegetation

The C content of marsh vegetation varied in a narrow range, with an average of 40.9 ± 2.8 %. *Cyperus madagascariensis,* which covers more than 50 % of the marsh area (Lammers et al., 2015), showed a clear C4 signature ($\delta^{13}$C: -13.2 to -12.4 ‰; Table 4). In contrast, *Argyreia vahibora,* which covers ~30 % of the marsh area, had $\delta^{13}$C values consistent with its C3 metabolism (-29.4 to -29.0 ‰), other marsh plant species showed $\delta^{13}$C values ranging between -29.5 and -24.1 ‰. $\delta^{13}$C values of terrestrial C3 and C4 vegetation in the catchment were previously reported as -32.5 ± 1.7 ‰ and -12.8 ± 0.6 8 ‰, respectively, but noting that the C3 values were likely biased by the understorey effect, with overall C3 vegetation in the region expected to show values closer to -29 ‰ (Razanamahandry et al., 2022). The latter would also be consistent with forest topsoil $\delta^{13}$C data between -28 and -26 ‰, while grassland topsoil $\delta^{13}$C in the catchment were higher and more variable (-23 to -16 ‰, Razanamahandry et al., 2022).

Table 4: δ¹³C values of marsh vegetation (leaves) found in Lake Alaotra sampled on February 2019 (rainy season)

| Scientific Name | δ¹³C |
|---|---|
| *Cyperus madagsacariensis* | -13.2 to -12.4 |
| *Cyclosorus gongylodes* | -29.5 |
| *Argyreia bahibora* | -29.4 to -29.0 |
| *Polygnum glabrum* | -29.5 |
| *Nymphaea sp.* | -24.1 |

## 4 Discussion

### 4.1 Sources of POC in the Lake Alaotra system

TSM concentrations of the inflowing water varied widely with season, with higher values during the rainy season (average $131 \pm 262$ mg L$^{-1}$) than during the dry season ($61 \pm 39$ mg L$^{-1}$). Sediment loads of inflowing water are within the range of TSM concentrations reported for streams and rivers of the Betsiboka basin (a basin which drains much from the grassland-dominated central highlands of Madagascar; Marwick et al., 2014), but higher than values reported from Rianala basin which drains part of the eastern slopes of Madagascar and is vegetated by low/mid-latitude humid evergreen forest (Marwick et al., 2014). There was a significant difference between TSM concentrations of inflowing water ($104 \pm 186$ mg L$^{-1}$) and lacustrine water ($18 \pm 13$ mg L$^{-1}$, Figure 9a), indicating that sedimentation must have occurred between the upstream area and the lake.

In addition, %POC was significantly lower in the inflowing rivers compared to lake and outflow waters (paired t-test, p-value < 0.001). Hydrodynamic sorting of sediments along the flowpath is one mechanism that could influence particle-associated POC and its characteristics including %POC (e.g., Bouchez et al., 2014), although its effect in our setting is difficult to predict given that we have no information on %POC in different grain size fractions. In general, however, we can anticipate that the finer material, with typically low POC content (<5%), is preferentially exported further downstream, while coarser material is retained within floodplains. While little empirical work on this has been done on how hydrodynamic sorting affects POC fluxes and characteristics in river systems, Repasch et al. (2022) found that the coarser mineral fraction and organic rich particles were more retained, while the fine mineral fraction become progressively more important in the Rio Bermejo (Argentina). Thus, we do not anticipate hydrodyanamic sorting to offer a viable explanation for the strong difference in %POC between inflowing rivers and Lake Alaotra.

In contrast to TSM, POC concentrations were higher in Lake Alaotra and its outflow compared to inflowing waters (Figure 8, Figure 9b). In the latter, POC concentrations were not strongly dependent on river discharge, while POC concentrations in the lake outflow increased steadily throughout the dry season – consistent with local inputs rather than with a link to POC derived from the catchment (Figure 7a; see discussion further below). δ¹³C-POC values measured in inflowing waters ($-24.6 \pm 1.8$ ‰) are largely consistent with δ¹³C values measured in subsoils within our study area catchment (Razanamahandry et al., 2022), although the relative contribution of POC to TSM (%POC) was on average higher than in soil profiles (Figure 10) which suggests that either direct vegetation inputs or more OC-rich soils closer to the stream network may contribute substantially (e.g. see Marwick et al. (2014) regarding the disproportional contribution of riparian zones). Along the aquatic continuum, %POC values increased further in the lake and its outflow (P-value for lake vs inflowing water and for outflow vs inflowing water: < 0.001) (Figure 7).

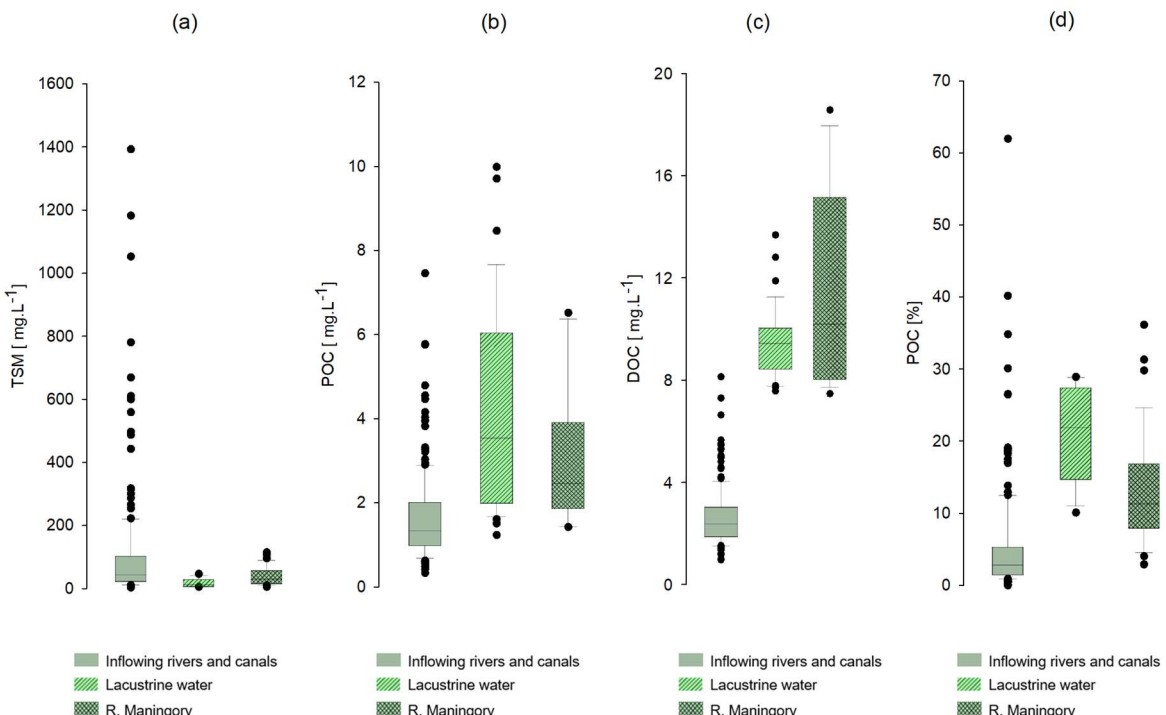

Figure 9: Boxplots of (a) TSM concentrations, (b) particulate organic carbon (POC) concentrations, (c) dissolved organic carbon (DOC) concentrations, and (d) the relative contribution of POC to the TSM pool of inflowing rivers and canals, lacustrine water and the outflowing River Maningory.

In addition to POC transported by the inflowing waters (i.e., from the upper catchment), there are two additional potential sources of POC to consider for Lake Alaotra and its outflow: POC formed along the river's path via wetlands (peat in marshes and aquatic plants) and *in situ* phytoplankton production within the lake. These new inputs of POC could be more important than riverine (terrestrial) inputs, due to the high productivity of the marshes and phytoplankton production. The fact that we observed higher POC concentrations as well as higher %POC in the surface water compared to inflowing water (Figure 10) indeed suggests that POC of the lake is to a large extent not derived from the river inputs, but must be linked to other sources such as phytoplankton biomass and/or marsh vegetation. Using the current distribution of different vegetation species in the marshes (Table 4), the expected $\delta^{13}C$ value of the mixture of different vegetation species based on their approximate relative abundance (Lammers et al., 2017) would be in the -21 to -18‰ range (~50% of -12.3 ‰, ~30% of -29.2 ‰ and ~20 % of ~-29.5 to -24.1 ‰). This value corresponds closely to $\delta^{13}C$-OC of the peat cores (-18.8 ± 1.4 ‰, see Table 3) and lake sediment cores (-18.5 ± 1.77 ‰, data not shown here), but is distinct from the $\delta^{13}C$ values of POC in the lake (-26.5 ± 2.1 ‰) and lake outlet (-25.3 ± 1.4 ‰) (Figure 10). This suggests that POC in the lake must be largely derived from other sources rather than from the remobilisation of OC from the marsh. The mean daily primary production rates we measured (0.5 ± 0.3 g C m$^{-2}$ day$^{-1}$) were moderate compared to the broader range of East African lakes, where primary production averages around 10 ± 5 g C m$^{-2}$ day$^{-1}$ (Morana et al., 2022), and similar to e.g. those measured in the oligotrophic lake Kivu (0.6 g C m$^{-2}$ day$^{-1}$; Darchambeau et al., 2014). The steady increase in POC concentrations during the dry season (Figure 7a) would be consistent with the development of phytoplankton biomass during a period when the lake water residence time increases. A widely used proxy for phytoplankton biomass is the chlorophyll a (Chl-a) concentration, and POC/Chl-a ratios (µg L$^{-1}$/ µg L$^{-1}$) in the water column can be used to evaluate the contribution of various sources of organic matter to POC in the lacustrine water (Cifuentes et al., 1988). Indeed, a high POC/Chl-a ratios suggests that organic matter is

primarily derived from terrestrial sources, while a low ratio implies that POC is derived from *in situ* phytoplankton production. Phytoplankton biomass has POC/Chl-a ratios between 40 and 200 whereas terrestrial organic matter POC/Chl-a ratios are typically higher than 500 (Gawade et al., 2018). The POC/Chl-a ratios of lacustrine water in our study ranged between 143 and 564 with an average value of 350±183.1. These values were relatively low and indicate that phytoplankton biomass represents an important fraction of the lake suspended POC. Moreover, the suspended organic matter of Lake Alaotra had POC/PN ratios close to those expected for phytoplankton (algae), between 7 and 9. The $\delta^{13}$C of phytoplankton ($\delta^{13}$C-Phyto) can be estimated based on $\delta^{13}$C-DIC data, but variability in isotope fractionation has been shown to be high. Indeed, an approximation of ~20 ‰ for the fractionation factor during C fixation by phytoplankton is often assumed (Peterson and Fry, 1987), many studies have found that the degree of isotope fractionation is determined by factors such as growth rates, cell geometry, nutrient status, and dissolved $CO_2$ concentrations (e.g. Burkhardt et al., 1999; Lammers et al., 2007b). Studies focussing on freshwater lakes have found inconsistent relationships between isotope fractionation and $pCO_2$ (Bade et al., 2006; de Kluijver et al., 2014), with $\varepsilon_p$ values (isotope fractionation between phytoplankton and $CO_2$) ranging between 1 and 16 ‰. $\delta^{13}$C-DIC values of Lake Alaotra water ranged between -9 and -4 ‰ (mean of 6.7 ‰) (Figure 11a), corresponding with $\delta^{13}$C-$CO_2$ values between -18 and -13‰ (mean of -15.7‰), which places our measured $\delta^{13}$C values of suspended POC ($\delta^{13}$C-POC) within the expected range for phytoplankton (-16 and -32 ‰) (Figure 11b). Thus, while the $\delta^{13}$C data alone do not unambiguously demonstrate that phytoplankton dominate the POC pool, several lines of evidence do point towards an important contribution from phytoplankton biomass: the much higher %POC than in inflowing rivers, low POC/PN ratios, relatively low POC/Chl-a ratios (particularly in May, Table 2), and lake $\delta^{13}$C-POC values do not match with marsh-derived organic matter (see Figure 10). No significant difference was observed between values of lacustrine water and outflow river during the rainy and dry seasons for TSM, POC concentration, $\delta^{13}$C-POC, and POC/PN values. Therefore, rather than exporting POC derived from soil erosion in the catchment, the Maningory river POC flux at the lake outlet appears to an important extent comprised of within-lake phytoplankton production.

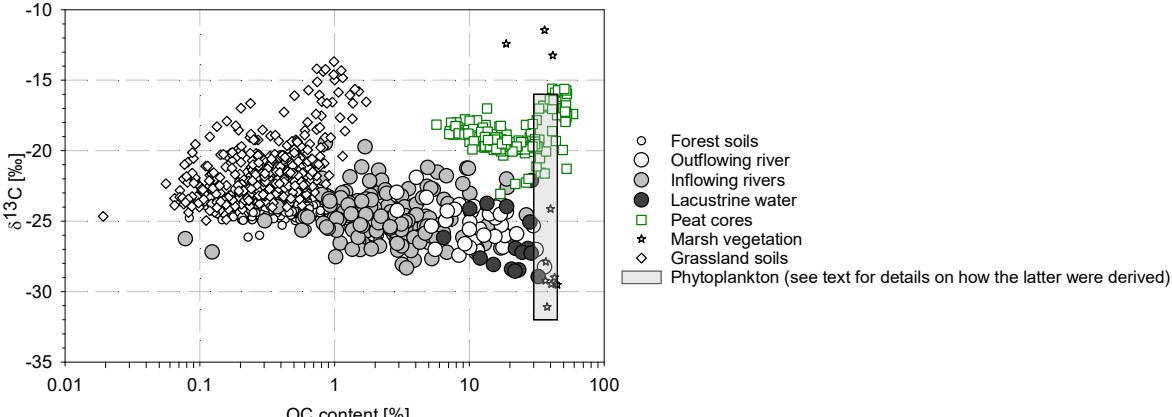

**Figure 10: %OC and δ¹³C-OC of marsh sediment cores, marsh vegetation, riverine suspended matter (inflowing rivers, outflowing river, and lacustrine water), soil samples (see in Razanamahandry et al., 2022) in the Lake Alaotra region, and estimated ranges for phytoplankton (see text for details on how the latter were derived).**

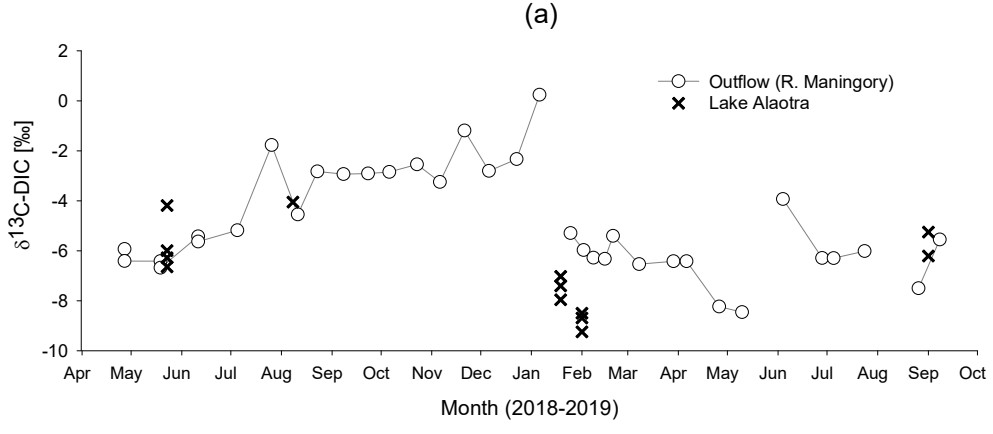

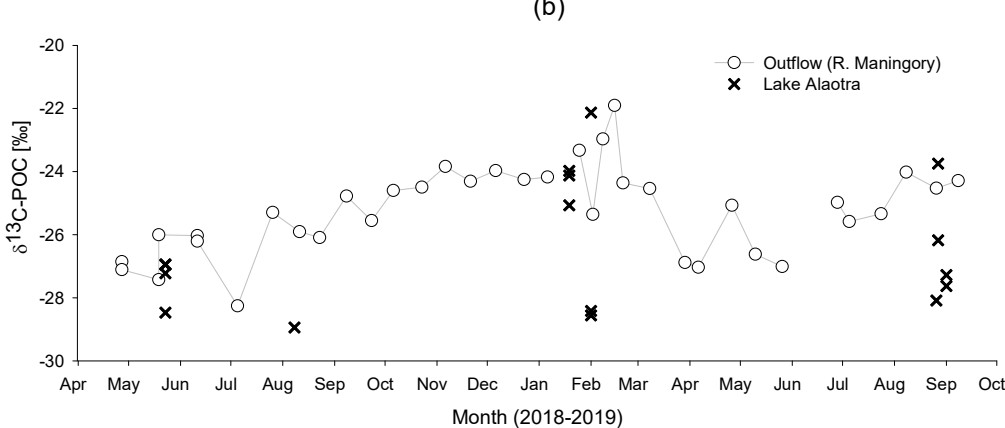

**Figure 11: (a) Seasonal variation of δ¹³C-DIC and (b) δ¹³C-POC of the Maningory River (lake outflow) and lacustrine water.**

### 4.2 Sources of DOC in the Lake Alaotra system

Lake Alaotra and its outflow were found to show consistently higher DOC concentrations than the inflowing water (Figure 9). Thus, analogous to the discussion on POC, we examine which possible sources are most likely to supply these new DOC inputs along the aquatic continuum. DOC concentrations of the inflowing water were relatively low ($2.6 \pm 1.1$ mg C L$^{-1}$) and show δ¹³C representing a mixture of C3-C4 vegetation (-23.0 $\pm$ 2.1 ‰) (Figure 8). These value are similar to those found in Marwick et al. (2014) in the Betsiboka river (central/western Madagascar; bulk DOC concentration ranges from 0.4 to 2.9 mg C L$^{-1}$ and δ¹³C-DOC varied between -29.5 and -15.4 ‰). In contrast, DOC in the Rianila basin (eastern Madagascar, largely covered by evergreen forest) was more depleted in ¹³C (δ¹³C values of -27.0 $\pm$ 1.2 ‰) but had a similar DOC concentration range ($2.6 \pm 1.4$ mg C L$^{-1}$; Marwick et al., 2014).

Possible sources for the net inputs of DOC within the lake include the production of DOC by phytoplankton, or DOC derived directly or indirectly from wetland vegetation. DOC production from phytoplankton has been shown to be rapidly mineralised in the water column and therefore does not appear to accumulate in the water column (Morana et al., 2014). Moreover, δ¹³C-DOC values within the lake and its outflow (-22.2 $\pm$ 1.0 ‰ and -21.4 $\pm$ 0.9 ‰, respectively) were relatively well constrained, and are distinct from those observed in POC which we expect to have a substantial phytoplankton contribution (see section on δ¹³C-POC). In addition, based on the relatively high seasonal variability in δ¹³C-DIC in the lake outflow, one would expect δ¹³C-DOC to show a correspondingly

high variability if phytoplankton production would fuel the net DOC inputs to the lake. Thus, while DOC and
POC are inherently operationally defined (i.e. by the filter pore size used in the sampling procedure), it appears
that the main sources of lake DOC and POC are strongly uncoupled (Figure 12). The net DOC inputs to the lake
are then likely the result of marshland vegetation inputs, which have been shown to be potentially important
sources of DOC to open water ecosystems (Lauster et al., 2006). Indeed, the $\delta^{13}$C-DOC values were close to $\delta^{13}$C-
OC values in our marsh sediment core profiles (-24 to -18‰; Figure 12: ), and to the values observed in marsh
vegetation, taking into account their relative abundance (see above). Thus, different lines of evidence point
towards marshland vegetation as the main source of the higher DOC concentrations in Lake Alaotra and its
outflow. While $\delta^{13}$C-DOC values in the lake and its outflow were relatively constant throughout the year, DOC
concentrations in lake waters were higher during the rainy season, when a higher flux of DOC from the OC-rich
marshes can be anticipated. While microbial mineralization of DOC has been shown to lead to possible shifts in
its $\delta^{13}$C values during the initial stages (Geeraert et al., 2016), such shifts are relatively modest (around 2 ‰ in
Geeraert et al., 2016, but less in previous studies cited therein) and are not anticipated to bias our interpretation.
DOC concentrations in Lake Alaotra are relatively high compared to DOC concentrations measured in a range of
East African lakes such as Lake Kivu, Edward and Albert (1.5–5 mg C L$^{-1}$, Morana et al., 2014, 2015) and in Lake
Victoria (1.2–3.6 mg C L$^{-1}$, Deirmendjian et al., 2020). This difference could be explained by the fact that Lake
Alaotra is a shallow system with a high relative area of surrounding wetland vegetation. A similar pattern has
been observed in Lake George, Uganda (Morana et al., 2022) which is also a very shallow lake fringed by
extensive *Papyrus* wetlands. Despite the high phytoplankton productivity in L. George, its surface waters showed
DOC values of 10–20 mg C L$^{-1}$, much higher than in rivers draining into the lake from savannah and rainforest (2
to 3 mg C L$^{-1}$). Similarly to what we observed in Lake Alaotra, $\delta^{13}$C data on DOC and POC pools in Lake George
demonstrate a strong decoupling of sources. In Lake George, $\delta^{13}$C-DOC values range from -27.5‰ to -25.0‰,
reflecting inputs primarily derived from surrounding wetland vegetation, whereas $\delta^{13}$C-POC values range from -
20.0‰ to -18.0‰, indicating dominance by aquatic primary production (Morana et al., 2022).The potential
importance of wetlands as a source of DOC in tropical lakes was also demonstrated for two shallow lakes in the
Congo Basin, Lake Tumba (average depth 3-5 m) and Lake Mai-Ndombe (average depth of 5 m, Borges et al.,
2022). Wetlands (flooded forests) surround both of these lakes, and leading to substantial DOC concentrations in
both Lake Tumba (14.7 mg L$^{-1}$) and Lake Mai-Ndombe (35.2 mg L$^{-1}$) (Borges et al., 2022).
The strong imprint of the surrounding wetlands on DOC inputs to Lake Alaotra is not mirrored in $CO_2$ and $CH_4$
concentrations, which do not show a marked increase between inflowing rivers and the lake proper (Figure 4, 6).
Here, outgassing (and/or oxidation in the case of $CH_4$) could explain the absence of a clear wetland imprint on
lake waters, yet some of the seasonality in $CO_2$ and $CH_4$ variations, in particular in the inflowing and outflowing
rivers also pointing towards a strong hydrological control modulating inputs from riparian wetlands. Indeed, the
moderate correlation between pCO$_2$ and dissolved $CH_4$ concentrations (Figure 5) suggests that the same processes
or environmental conditions drive the pCO$_2$ and $CH_4$ variations. A strong increase of pCO$_2$ and $CH_4$ was observed
during the rainy season in February (Figure 5) when both precipitation, water levels (Figure 6) and freshwater
discharge (Figure 2) increased – thus coinciding with an expected increased hydrological connectivity between
wetlands and rivers and inputs of $CO_2$ and $CH_4$ from riparian wetlands to rivers. As the water level rises, the river
floods riparian wetlands, where the decomposition of organic matter in wetland sediments enriches the water with
dissolved $CO_2$ and $CH_4$. When the water flows back from the wetlands into the river further downstream, it
increases $CO_2$ and $CH_4$ levels in the river, as observed in the Amazon (Richey et al., 2002), Zambezi (Teodoru et
al., 2015), and Congo (Borges et al., 2019) rivers. Riparian wetland inputs seem to rapidly reestablish high $CO_2$
and $CH_4$ levels in the outflowing river leading to values equivalent to the inflowing rivers, suggesting a very
limited imprint of the lake on the $CO_2$ and $CH_4$ emissions flowing waters. The relative importance of lentic $CO_2$
and $CH_4$ emissions compared to lotic $CO_2$ and $CH_4$ emissions is small based on extrapolation of field
measurements with spatial datasets at the scale of the Amazon Basin (Chiriboga et al., 2023), as well as the
continental scale of Africa (Borges et al., 2022). The lower $CO_2$ and $CH_4$ emissions from lentic systems compared
to lotic ones resulted from lower areal flux densities as well as a lower total surface area (Chiriboga et al., 2023;
Borges et al., 2022).

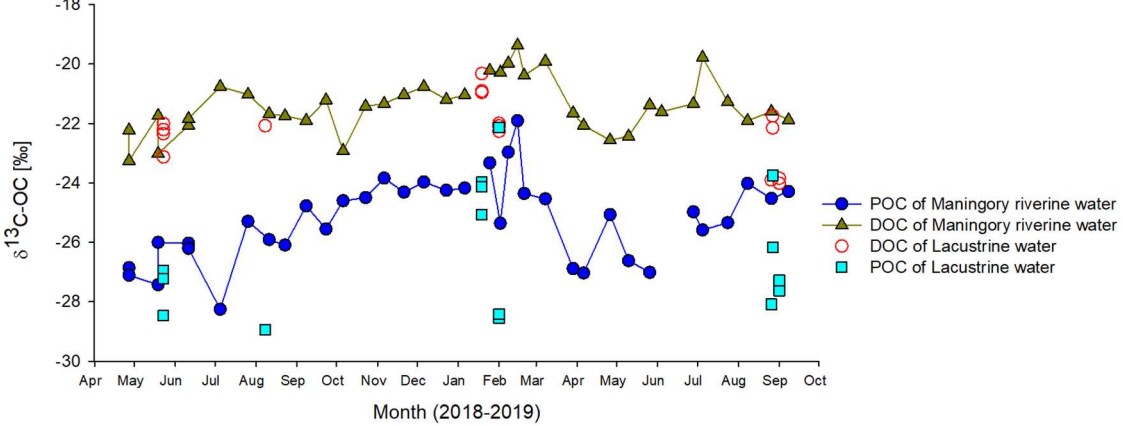


**Figure 12: Seasonal variation of $\delta^{13}$C-POC and $\delta^{13}$C-DOC in Lake Alaotra and in the Maningory River (lake outflow).**
**5 Conclusions**
We present a comparative dataset of physico-chemical and biogeochemical proxies measured in inflowing water,
lacustrine water, and the outflow of Lake Alaotra, focussing on tracing the main sources of C along the aquatic
continuum and how the lake and surrounding wetlands affect dissolved and particulate OC pools. Our results
highlight the dominant role of riparian wetlands in controlling $CO_2$ and $CH_4$ dynamics, with hydrological
connectivity driving seasonal variations in GHG inputs to the river system. The lake itself has a limited influence
on $CO_2$ and $CH_4$ emissions as wetland contributions rapidly reestablish high concentrations in outflowing waters.
This pattern reflects the broader significance of wetland-river interactions in shaping carbon cycling and
greenhouse gas fluxes in tropical river systems. Furthermore, our data show that the suspended sediment load
derived from the upstream catchment is largely lost before it enters Lake Alaotra, and that the concentrations of
DOC and POC in the lake and its outlet were much higher than in the inflowing waters. POC/Chl-a ratios of
lacustrine water were low, and the POC/PN ratios and $\delta^{13}$C-POC values of the lacustrine suspended organic matter
were consistent with a strong contribution by phytoplankton production. In contrast, $\delta^{13}$C-DOC values within the
lake and outflow were consistently higher than $\delta^{13}$C-POC, and the surrounding marshes appeared to be the primary
source of the lacustrine DOC inputs. This study indicated that Lake Alaotra is highly dynamic in terms of organic
C, and acts as an active hotspot (sensu McClain et al., 2003) in terms of modifying C fluxes and sources along the
aquatic flowpath. *In situ* production and marshes are the primary sources of organic carbon in the lacustrine water
column. The findings of this study are crucial for interpreting lake sediment archives and for tracing sediment
mobilization from the eroded landscape in the Lake Alaotra Region.

## 6 Data availability

The full dataset generated in this study can be found as an electronic supplement. This supplement includes a number of ancillary measurements on proxies that are not discussed in this paper (stable hydrogen and oxygen isotope data of surface water samples and major element concentrations), but which we have kept in the database so that potential users of our data have access to these additional parameters. The methodology for these measurements is briefly described in the Supplementary information.

## 7 Authors contributions

G.G and S.B. designed the study project with contributions of L.B and V.F.R. TaR. and ToR. co-supervised the project and fieldwork in Madagascar. L.B. and V.F.R. planned fieldwork and collected samples. V.F.R. conducted the main sample analyses and led the manuscript writing with S.B., C.M, and A.V.B. who also contributed to $CH_4$ and TA analyses. All authors contributed to data interpretation and manuscript revisions.

## 8 Competing interests

S.B. is co-editor-in-chief of Biogeosciences.

## 9 Acknowledgements

This research is part of the MaLESA (**Ma**lagasy Lavaka, **E**nvironmental reconstruction and **S**ediment **A**rchives) project funded by KU Leuven (Special Research Fund). Travel and research grants were provided by YouReCa, and FWO (11B6921N, 12Z6518N, V436719N). We thank Zita Kelemen, Lore Fondu, Christophe Coeck, David Soto, Elvira Vassilieva, Jasper Verdonck and Yannick Stroobandt (KU Leuven, Leuven, Belgium), Marc-Vincent Commarieu and Jean-Pierre Descy (Université de Liège, Liège, Belgium), Marie Paul Razafimanantsoa (LRI, Antananarivo, Madagascar), for technical and analytical assistance. We thank ONG Durrell, and MEDD (Ministère de l'Environnement et de Development Durable) of Madagascar for authorizing sample collection in the protected area of the Lake Alaotra. A.V.B. is a Research Director at the Fonds National de la Recherche Scientifique (FNRS, Belgium). We are grateful to Peter Douglas and Dailson Bertasolli for providing constructive and insightful reviews on an earlier version of this manuscript.

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
