# Peer review of "Biogeochemical functioning of Lake Alaotra (Madagascar): a"

_EGUsphere, 2024_

## Author Comment (AC1)

We would like to thank Peter Douglas and Dailson Bertassoli for the time and effort taken to provide constructive and insightful comments on our manuscript. Below, we have copied their original suggestions and comments in full, followed by a response to each of the issues raised and how these will be addressed in a revised version. Should we be invited to revise the manuscript, a more detailed reply detailing the actual changes to the manuscript will be provided subsequently.

**RC1**: 'Comment on egusphere-2024-2213', Peter Douglas, 05 Sep 2024

*Ref: This is an interesting article that tackles an important but understudied topic, namely the role of (tropical) lakes in modulating inland water carbon fluxes. I'm glad to see the focus on carbon sources and fluxes upstream and downstream of the lake. There is a huge wealth of data contained in the paper, and for the most parts the authors do a solid job of analyzing these data coherently. Ultimately I think it will be a valuable contribution.*

*However, there are a number of issues that need to be addressed before this should be published in Biogeosciences. I would characterize these as major revisions. I summarize major issues below, and then provide line by line comments.*

*Peter Douglas*

***Introduction:*** *I found the introduction to be somewhat unfocused, and a clear rationale for the study does not emerge. I think the authors should put more attention to the importance of this study in a global context, and how this site is representative of globally important ecosystems. What questions in global carbon cycling and inland water carbon fluxes does this study help us address? The discussion of environmental impacts of sediment from soil erosion is interesting, but given the topic of this study it seems the introduction should be more focused on carbon cycling. Also, it seems part of the expressed motivation of the study is to interpret paleo-proxy data, which I think is tangential. Again I think these data are more interesting in terms of modern day carbon cycling and that should be a larger focus. See cited paper Cole et al., 2007 and Regnier et al., 2020 for examples of key questions in inland water carbon cycling.*

*Regnier, Pierre, et al. "The land-to-ocean loops of the global carbon cycle." Nature 603.7901 (2022): 401-410.*

> Reply: We appreciate this clear request to reformulate the context and objectives – this also falls in line with similar suggestions by Ref#2. We can certainly focus more here on the carbon cycling aspects and global relevance of (sub)tropical lakes; as this is indeed likely what many readers will be interested in. Nevertheless, we do wish to point out that we do not consider the importance of this study in terms of paleo-proxy data interpretation as tangential – this was in fact the main driver of conducting this work. In the broader framework of the project which this study is part of, we examine various proxies in floodplain, marsh and lake sediment cores – and any interpretation of these requires a basic understanding of the

biogeochemistry of the lake itself, and a solid characterization of the inputs and outputs. A manuscript is being finalized on those sediment data, and the conclusions from the current study are an important element guiding the sediment proxy interpretations. Hence, we will on the one hand better explain this part of the rationale, while at the same time better express the global inland water C cycle context.

**Ref: Sampling Design:** *I think the selection of sampling sites and the sampling design needs more clarification and elaboration. How were inflowing rivers chosen and why were some sites sampled at a higher frequency? How were the lake sampling sites chosen, and importantly what depth was water sampled from? Lake sampling is not really described at all. How far downstream of the lake outlet was the outflowing river sampled? It seems to me the marsh sampling should be discussed in the same section as the water sampling, or at least before the analytical methods. Why were only marsh plants sampled and not terrestrial plants or phytoplankton, which are also probably important OM sources? The point of the cores from the marshes is not that clear- maybe explain a bit more the rationale for collecting cores as opposed to surface samples, and what information is provided by the depth profiles.*

Reply: We will add details providing all relevant information on sampling in the revised version. Briefly, inflowing rivers were sampled along the entire lake area (when there was sufficient water flow), with a selection made for regular sampling based on their accessibility for collaborators on site and that these would have flow year-round. Lake sampling took place along a N-S gradient covered by boat, all water samples were taken with a Niskin bottle from the surface (approx 0.5 m depth).

**Ref: Analytical Methods:** *This section is very dense and hard to follow. I would suggest splitting into subsections if journal allows. Some reorganization is needed. PCO2 measurements should proceed the estimates of DIC concentration since they are used for that. There is no mention at all of the CH4 measurements that are presented, which is a major oversight.*

Reply: We will bring more structure into the Methods section as suggested. $CH_4$ methodology will be added.

**Ref: Results:** *This section is very dense and hard to follow. If possible to break into subsections that would make reading easier.*

Reply: As for the section on analytical methods, we will break this up into subsections as suggested.

**Ref: Vegetation data:** *As mentioned above, having d13C values of local terrestrial plants and phytoplankton, or at least estimates, would really be helpful and complement the data shown in Table 3. The 13C values of subsoils are referenced but the actual numerical values*

*should be summarized, and it is unclear why topsoils are not included. In line 448 there is discussion of a mixing model based on plant isotopic values, but this is quite unclear and hard to follow. More details are needed, and the relative abundance of different plant types should be specified, ideally in a table.*

Reply: Samples of terrestrial vegetation were taken and reported in Razanamahandry et al. (2022, https://doi.org/10.5194/bg-19-3825-2022). We will refer to this in the revised version. Topsoil data (from the same study) can also be cited again. Regarding phytoplankton, we sampled the total suspended matter pool in the lake (as discussed in the next comment); but see no way of sampling pure phytoplankton in such settings. As discussed in the manuscript, suspended matter in lake surface water seemed to be mainly composed of phytoplankton.

**Ref: *Phytoplankton 13C calculation:*** *I'm afraid this is a gross oversimplification. This fractionation factor can very hugely and is sensitive to biological and environmental variables like growth rate and light, as well as differences between taxa. Use of a single value of 20‰ is too simplistic, and based on my quick review is on the high end, as opposed to an average value. So I don't think this is appropriate. In addition the uncertainty in this estimate needs to be accounted for. There is an extensive literature on this, but see for example:*

*Burkhardt, Steffen, Ulf Riebesell, and Ingrid Zondervan. "Effects of growth rate, CO2 concentration, and cell size on the stable carbon isotope fractionation in marine phytoplankton." Geochimica et Cosmochimica Acta 63.22 (1999): 3729-3741.*

*De Kluijver, A., Schoon, P. L., Downing, J. A., Schouten, S., & Middelburg, J. J. (2014). Stable carbon isotope biogeochemistry of lakes along a trophic gradient. Biogeosciences, 11(22), 6265-6276.*

*The Fry et al citation is quite outdated. Including more uncertainty in this calculation is required, and likely has effects on the inferred contribution of phytoplankton.*

Reply: We fully agree that our approach is a simplification, and indeed there is a wealth of knowledge on isotope fractionation in phytoplankton and the controlling factors. However, since we have no information on the phytoplankton community composition, we cannot make full use of this knowledge and are, in our opinion, limited to a coarse estimate. However, we will look into the literature to screen for more recent approaches used by others, and include an uncertainty evaluation on the phytoplankton $\delta^{13}$C estimates by using a range of likely values of fractionation factors compiled from literature.

**Ref: *Consideration of hydrodynamic processes:*** *Differences in %POC between the TSM and soils was suggested to indicate a different, more organic rich source. However, this could also reflect sorting and selective settling of eroded soil that could lead to greater %POC in the lake. For example, soil minerals may selectively settle and deposit in*

*sediments, whereas OM rich material may be more likely to be suspended. I think this is worth consideration. This could also potentially explain the higher %POC in the lake vs the inflowing rivers.*

> Reply: We will add some literature and discussion on this – indeed important-aspect. One would expect, however, that the fraction ultimately transported to the lake itself would be the finer clay fraction – which typically has %OC lower than 5%, thus substantially lower than those observed in lake suspended matter.

***Ref: More detailed implications:*** *I think it would be really valuable to see a bit more discussion of the implications of this work at the end of the discussion. What does this mean for the role of lakes in riverine carbon transport and emissions? Does the presence of lakes in (tropical) rivers lead to a net increase or decrease in emissions, and does it change the overall source of C that is being respired? Do your results have implications for the carbon cycle effects of current anthropogenic changes in the catchment and the lake? Will greater erosion and/or rice production lead to changes in the GHG fluxes from this system or the downstream export of carbon? I think addressing some of these questions will enhance the impact of the paper.*

> Reply: This suggestion is in line with earlier suggestions on the introduction (context & objectives), so indeed it would be logical to come back to this in the Discussion. Note that some of the questions raised can only be addressed broadly and might be somewhat speculative, but we see the use of raising the questions and providing some pointers. We will refer to a recent synthesis by our group (Borges et al. 2022, https://doi.org/10.1126/sciadv.abi8716) that showed that emissions from lentic systems were marginal to lotic systems at the scale of the African continent. The same seems to hold from the Amazon basin (Chiriboga et al. 2023, https://doi.org/10.1007/s00027-023-01039-6). Both studies are based on field measurements upscaled at larger scales with GIS so provide quantitative estimates albeit admittedly uncertain.

*Ref: Line by line comments:*

*Line 38: this phrase about degassing is not clear to me.*

> Reply: This will be reformulated – we also expect degassing of $CO_2$ once the riverine water (oversaturated in $CO_2$) enters the open lake waters, where higher turbulence due to wind fetch increases the gas transfer velocity.

*Ref: L61: This sentence is not clear. If the C fixed is not buried or emitted where does it go? Exported as DOC downstream?*

> Reply: This will be corrected – should have read "20 times higher than organic C burial" – thus indicating fast recycling of phytoplankton biomass.

*Ref: L70: Be more clear why data from Madagascar is valuable in a global sense.*

Reply: We will expand on this aspect.

*Ref: L90: Specify how the wetlands have been altered.*

Reply: We will elaborate on this in the revised version, but the main alteration is conversion to rice paddies.

*Ref: L105: This is an important point and isn't totally clear here. Based on the Broothaerts paper there is a huge amount of sedimentation in the floodplain (100x greater than the lake) and a lot in the wetland (10x greater than the lake), leading to very low sedimentation in the lake itself.*

Reply: We will expand this section. It is indeed an important aspect that we come back to in the Discussion, and in a follow-up paper that discusses our elemental and stable isotope data in floodplain, marsh, and lake sediment cores in which our data suggest that sediments from hillslope erosion have been trapped mostly in the floodplain and marshes.

*Ref: L130: This sentence is redundant with the earlier part of the paragraph.*

Reply: These will be merged.

*Ref: Figure 2: Is it possible to add how this hydrological difference affects lake level?*

Reply: We will verify if historical lake level data can be found; otherwise see Figure 3 where water levels for the outflowing river (Maningory) are shown for 2018-2019; this should be a reasonable proxy for lake water levels (and nicely matches the longer-term outlet discharge in terms of timing and shape).

*Ref: L176: Awkward phrasing- try to rewrite.*

Reply: Point taken, we will rephrase this.

*Ref: L195: Much more details on lake sampling needed.*

Reply: More details will be added in the revised version.

*Ref: L298: There is not really much on d15N in the paper. Was it used at all? I recognize the authors are providing a wide range of data that was not used in the paper. Perhaps methods for these analyses should be in a supplement so that they can be used later but do not distract in the main text.*

Reply: Indeed, we found no clear use for them in this manuscript but feel it important to put the $\delta^{15}N$ data out (as well as some other parameters), as they

might still prove to be useful for other researchers e.g. for data syntheses or meta-analyses. We will modify the Methods to explicitly mention the methodology (measured in the same run along with OC, PN, and $\delta^{13}C$).

*Ref: Figure 6: The cause of the gaps in data are not clear to me, make clearer here or in the methods.*

Reply: $pCO_2$ was measured in situ (LICOR-820, headspace equilibration) during our own field campaigns, but not by the team performing the regular monitoring for logistical reasons. CH4, in contrast, was analysed on discrete samples preserved on site and measured later in the lab – hence, for this parameter a more complete dataset is available. For logistical reasons, not all of the rivers sampled during our own field campaigns were monitored regularly, this was only the case for a sub-set of rivers.

*Ref: L358: It is interesting that %POC is lower in the outflow than the lake. What is the source of inorganic TSM in the outflow?*

Reply: This is an interesting point – we will look into this more closely. Note however, that the Maningory data (outflow) include data from regular sampling throughout the year, while the lake samples were taken during a short time periods of the full field campaigns. The CH4 data indicate lateral inputs from riparian wetlands to the Maningory, which might be one possible hypothesis.

*Ref: Figure 9: Is any information on the age-depth relationships in the cores available. The core data in general is not that informative, so maybe it could be more simply summarized as a source of OM.*

Reply: Yes, $^{14}C$ dating was performed on the M3 core, these are reported in Broothaerts et al. (2022); we will add these data in the text of the revised version. We will carefully weight the options to summarize the core data or keep its current presentation form (the full depth profile) – see also last comment by Ref#2 who asks whether some interpretation of the depth trends can be provided.

*Ref: L454: Give numerical values (i.e. average plus standard deviation) for the lakes in East Africa being compared to.*

Reply: These data will be summarized and added in the revised version.

*Ref: L469: In addition to the issues discussed above, if there is high C fixation this can lead to enrichment of d13C in the water column (i.e. a Rayleigh distillation effect) and potentially lead to erroneous estimates based on the fractionation factor.*

*See for example: Van Dam, Bryce R., et al. "CO2 limited conditions favor cyanobacteria in a hypereutrophic lake: an empirical and theoretical stable isotope study." Limnology and Oceanography 63.4 (2018): 1643-1659.*

> Reply: In line with the earlier comment on isotope fractionation for phytoplankton, we will re-examine this aspect of the study and include a more critical discussion of factors influencing isotope fractionation between DIC and phyto biomass.

*Ref: L474: This begs the question: are the differences between the inflow water and the lake/outflow water significant?*

> Reply: Yes, indeed – for some reason we have not mentioned this explicitly; we will do so in the revised version and will consider an additional graph in the supplement whereby the seasonal data from inflowing rivers and outflow are plotted jointly.

*Ref: L513: Again, provide numerical values for these other lakes.*

> Reply: will be included in the revised version.

*Ref: L517: Need a citation for these data from the Congo.*

> Reply: Reference will be added.

*Ref: L524: I think a more detailed explanation for the connectivity causing high pCO2 and pCH4 is needed.*

> Reply: We will expand this section of the Discussion in the revised version.

**RC2**: ['Comment on egusphere-2024-2213'](), Dailson Bertassoli, 16 Oct 2024

*Ref: General comments:*

*This study investigates the carbon biogeochemistry of Lake Alaotra, Madagascar, by analyzing variations in carbon pools, $CO_2$, $CH_4$, and other parameters over a complete hydrological cycle. It offers valuable and much-needed data that contribute to advancing discussions on the role of tropical lakes as "biogeochemical reactors." While the authors have made an important effort to underpin their discussion, I believe that some broad generalizations weaken certain key findings. Therefore, I believe this manuscript should undergo major revision or be resubmitted before publication.*

*The introduction offers extensive details about the study area but lacks sufficient emphasis on the main research objectives, making it difficult to fully understand the rationale of the study. The authors should reconsider the level of attention given to 'lavakas' (l12, l83–l86, l124) and paleoenvironmental interpretations (l108–112), as these topics are not directly connected to their main findings. Additionally, the manuscript would benefit from clarifying the gaps in the carbon cycle that this research aims to address, providing a stronger rationale for the study.*

> Reply: This suggestion is in line with those of Ref#1, see response there – we will re-organize parts of the Abstract and Introduction (and Discussion) to frame the study more in the context of global/regional C cycling and the role of (sub)tropical lakes. The paleo-environmental proxy context was and remains an important justification for us to have conducted this work, but we will express this better and place it more in the background and link to published and submitted companion papers; we understand the link might not be obvious in the current manuscript.

*Ref: The authors mention a "selection of rivers," but the criteria for choosing these two rivers and their representativeness regarding the overall water balance of Lake Alaotra are not clearly explained. Additionally, highlighting these rivers on the map in Figure 1 would improve visualization. There is also some ambiguity about how many rivers were sampled during the high- and low-water field trips. Overall, the sampling scheme and the methodologies used are somewhat unclear and lack key details. For example, what were the water depths at the sampling sites? How were the sampling points in the lake selected? While coordinates are provided in the Supplementary Data, important implications regarding the sampling strategy should be discussed in the main text. For instance, $CH_4$ concentrations in lakes can vary significantly depending on proximity to the margins. Addressing these aspects would greatly enhance the study.*

> Reply: These are mostly in line with suggestions from Ref#1 and will be taken into account: more details will be added in the Materials and Methods section, and rivers where sampling was done regularly will be indicated with a separate symbol on Figure 1.

*Ref: The results section would be much more reader-friendly with the inclusion of additional graphs and tables. Biplots, in particular, could greatly aid in comparing the ranges of organic carbon across soil, lake, and river samples. In the discussion, although the evidence suggesting that the increase in %POC in the lake is likely driven by phytoplankton input is relatively solid, the text is structured in a way that makes the argument somewhat unclear. A clearer and more focused presentation of this idea would improve the coherence of the discussion. Also, the authors may get interesting perspectives on erosional patterns by delving deeper into the TSM changes.*

> Reply: Thanks for this suggestion. We will work on some additional plots and consider whether they would fit best in the main text or in the Supplement. In line with comments from Ref#1 on discussing possible changes in %OC during transport of suspended sediments, we will work to improve the line of reasoning to the conclusions in the importance of phytoplankton within the lake.

*Ref: My main concern, however, lies in section 4.2. It is important to emphasize that the separation between POC and DOC fractions, based on size, represents an "instrumentalist" approach that overlooks key factors. Most importantly, a primary control on %POC and TSM, the energy of the environment, was severely neglected in the discussion. This, naturally, has significant implications for some of the interpretations. Similarly, degradation, which is also size-dependent, probably play a critical role in shaping the OM δ13C signatures observed in the system but was not adequately addressed. Additionally, although the conclusion that DOC and POC sources are uncoupled is reasonable, the entire discussion regarding DOC sources seems to oversimplify the system and is not fully supported by the data presented in the article.*

> Reply: The distinction between POC and DOC is indeed operationally defined. We assume the comment on the energy of the environment is similar to the comment of Ref#1 regarding hydrodynamics influencing particle composition (deposition, resuspension, particle sorting, etc.) -see our reply there. Degradation may indeed have small effects on $\delta^{13}$C – we can include this into the Discussion of the revised version although we do not see that this will fundamentally change our conclusions. However, we are unsure what the reviewer refers to with the statement that the discussion on DOC sources is not fully supported by the data presented – some more details would have been welcome. Either way we will revisit and amend the relevant sections critically.

*Ref: Lastly, it would be great if the authors could place the obtained results within a broader regional context, highlighting their implications for the current understanding of the tropical carbon biogeochemistry cycle.*

> Reply: This suggestion is in line with those of Ref#1, see response there – we will re-organize parts of the Abstract and Introduction (and Discussion) to frame the

study more in the context of global/regional C cycling and the role of (sub)tropical lakes. The paleo-environmental proxy context was and remains an important justification for us to have conducted this work as well as for submitted or published companion paper, but we will express this better and place it more in the background.

*Ref: Specific comments:*

*L12-14: As I read these lines, I thought the main focus of the article to be different. Consider focusing the beginning of the abstract towards the main target of this article.*

Reply: See response to previous comment.

*Ref: L26: Not necessarily 'surprising'*

Reply: It was surprising to us. The vast majority of lakes we have studied so far show little or no change in DOC concentrations (and $\delta^{13}$C) from the inflowing rivers. See Discussion on lines 504-517 in the initial version of the manuscript. Unless this is considered a critical point, we would prefer to leave this as it is.

*Ref: L31: I'm not sure if "was expected" is the right expression here. Do you mean "was found," as in: "δ13C data indicated that marsh vegetation was the main source of net DOC inputs, while phytoplankton likely contributed to POC in the lacustrine waters."?*

Reply: Indeed, bad choice of words: we will rephrase.

*Ref: L100: I recognize that this study is present in the literature, but its conclusions don't seem well-supported by the findings. I'm not sure if continuing to reference it truly benefits the advancement of science.*

Reply: Fair point – we indeed agree that their conclusions are not supported by subsequent studies (and we feel this is clear from the rest of the sentence and the following one); but we can for sure consider to remove it, or to rephrase to make it more clear that this is a disputed claim.

*Ref: L108-L112: I am not sure how this directly relates to the main subject of the article.*

Reply: We feel this is an important justification or aspect of the context of this paper: the lake is situated in a region that has experienced major anthropogenic disturbance and deforestation, and as stated in the introduction one of our aims is also to use lake sedimentary records to reconstruct some of these changes. We would thus prefer to keep this sentence.

*Ref: Figure 1: Please consider highlighting which rivers were measured monthly and which were not measured during the dry season.*

Reply: We should be able to accommodate this, by using a separate symbol for the rivers that were measured monthly.

*Ref: Figure 9: Are there any considerations regarding changes in the parameters for different depths that can contribute to your discussion?*

Reply: We acknowledge that the data in Figure 9 are currently not discussed in detail: that will be part of a broader manuscript in preparation that discusses similar data from across the floodplain-marsh-lake gradient; and we mainly presented them here since they show %OC and $\delta^{13}$C values from what we consider to be a potentially important 'end-member' source of OC to the lake. Ref#1 suggested to replace this Figure with a summary, rather than present the full profile. We feel there are good arguments for both options (summarize the data, *versus* keeping the full profiles but add some more interpretation/discussion), and will carefully weight these during our revisions.

---

## Author Response (AR1)

We would like to thank Peter Douglas and Dailson Bertassoli for the time and effort taken to provide constructive and insightful comments on our manuscript. We also to thank the Editor for detailed feedback and for relaying the constructive comments and suggestions from Drs. Douglas and Bertassoli. We appreciate the time and effort you and the reviewers have devoted to evaluating our manuscript and for recognizing the significance of our work on the understudied ecosystems in Madagascar.

Below, we have copied their original suggestions and comments in full, followed by a response to each of the issues raised and how these are addressed in our revised version. As per your instructions, we provide a marked-up manuscript clearly identifying the changes made and include a detailed point-by-point response to all comments with references to line numbers in the revised text.

*Editor comments:*

*EC: Title: I assume that you meant "land-ocean (aquatic) continuum" by "land-ocean gradient". Given the usual usage of the land-ocean gradient to indicate something like haline and thermal changes, would a more common term "continuum" better fit your intention?*

> REPLY: We agree - we have modified the title accordingly, and have also checked/changed the use of 'gradient' throughout the manuscript, and modified where appropriate. **Line 2 (clean and revised versions)**

*EC: Abbreviation Usage: Consistently define and use abbreviations such as "Chl-a," "DOC," and "POC" across the abstract and main text.*

> REPLY: This has been checked throughout, and modified.

*EC: Phytoplankton and POC Discussions: Revise the text to avoid unsupported suppositions about phytoplankton as a source of POC.*

> REPLY: See replies to different referee comments below: we have more carefully phrased our arguments for phytoplankton contributions to the POC pool and have toned down the $\delta^{13}C$ arguments for their importance based on a more critical look at the variability of isotope fractionation in phytoplankton. **Section Abstract: Lines 30–32 (Clean version) and Lines 35–37 (revised version)**
> **Section Discussion: Lines 563–576 (Clean version) and Lines 648–663 (revised version)**

*EC: Floodplain and Rice Field Coverage: Update the Abstract to include mention of the large coverage of floodplains and rice fields, and revise Figure 1 to better distinguish areas between wetlands and rice fields.*

REPLY: The Abstract has been updated to mention floodplains and rice fields, and Figure 1 revised to clearly distinguish wetlands from ricefields as suggested.
**Section Abstract: Line 15 (clean and revised versions)**
**Figure 1:  Line 179 (clean version) and Line 230 (revised version)**

*EC: Phytoplankton 13C Calculation: Consider the suggested approach by Campeau et al. (2017) to improve the discussion on source partitioning and ensure alignment with recent methodological advances in the field.*

REPLY: We assume that reference is made to:

Campeau, A., Wallin, M.B., Giesler, R. et al. Multiple sources and sinks of dissolved inorganic carbon across Swedish streams, refocusing the lens of stable C isotopes. Sci Rep 7, 9158 (2017).

While an interesting dataset, this reference in our opinion makes use of the Keeling/Miller-Tans approach under settings where it is not valid to do so, and we see no merit in propagating this invalid use in aquatic systems. It also offers to solution to better estimates of the phytoplankton $\delta^{13}C$ end-member. We do however agree with the importance of improving the latter, and/or to discuss more critically and put some measure of uncertainty on this. We now expanded upon this discussion, and refer to Bade et al. (2006), Lammers et al. (2017), Burkhardt et al. (1999), and de Kluijver et al. (2014) – see replies further on for more details.
**Section discussion:   Lines 563–576 (clean version) and Lines 648–663 (revised version)**

*EC: CO2 and CH4 Sources: Expand the discussion to incorporate potential contributions from lake sediments and provide a more balanced view of CO2 and CH4 sources within the lake system.*

REPLY: The discussion has been expanded upon, as requested.
**Section discussion: Lines 640–652 (clean version) and Lines 733–744 (revised version)**

*EC: Figures: Correct month abbreviations and ensure accuracy and consistency across all figures.*

REPLY: Thank you for pointing this out. We have corrected the month abbreviations across all figures to ensure consistency and accuracy. Standard three-letter abbreviations have been used throughout, and we have verified that the labels correspond correctly to the data represented in each figure.
**Figure 2: Line 191 (clean version) and Line 254 (revised version)**
**Figure 3: Line 386 (clean version) and Line 454 (revised version)**

**RC1**: Peter Douglas, 05 Sep 2024

Ref: *This is an interesting article that tackles an important but understudied topic, namely the role of (tropical) lakes in modulating inland water carbon fluxes. I'm glad to see the focus on carbon sources and fluxes upstream and downstream of the lake. There is a huge wealth of data contained in the paper, and for the most parts the authors do a solid job of analyzing these data coherently. Ultimately I think it will be a valuable contribution.*

*However, there are a number of issues that need to be addressed before this should be published in Biogeosciences. I would characterize these as major revisions. I summarize major issues below, and then provide line by line comments.*

*Peter Douglas*

**Introduction:** *I found the introduction to be somewhat unfocused, and a clear rationale for the study does not emerge. I think the authors should put more attention to the importance of this study in a global context, and how this site is representative of globally important ecosystems. What questions in global carbon cycling and inland water carbon fluxes does this study help us address? The discussion of environmental impacts of sediment from soil erosion is interesting, but given the topic of this study it seems the introduction should be more focused on carbon cycling. Also, it seems part of the expressed motivation of the study is to interpret paleo-proxy data, which I think is tangential. Again I think these data are more interesting in terms of modern day carbon cycling and that should be a larger focus. See cited paper Cole et al., 2007 and Regnier et al., 2020 for examples of key questions in inland water carbon cycling.*

*Regnier, Pierre, et al. "The land-to-ocean loops of the global carbon cycle." Nature 603.7901 (2022): 401-410.*

> Reply: We appreciate this clear request to reformulate the context and objectives – this also falls in line with similar suggestions by Ref#2. In the revised introduction, we focus more on the carbon cycling aspects and global relevance of (sub)tropical lakes. Nevertheless, we do wish to point out that we do not consider the importance of this study in terms of paleo-proxy data interpretation as tangential – this was in fact the main driver of conducting this work. In the broader framework of the project which this study is part of, we examine various proxies in floodplain, marsh and lake sediment cores – and any interpretation of these requires a basic understanding of the biogeochemistry of the lake itself, and a solid characterization of the inputs and outputs. A manuscript is being finalized on those sediment data, and the conclusions from the current study are an important element guiding the sediment proxy interpretations. Hence, we will on the one hand better explain this part of the rationale (this remains part of the introduction, but does not feature anymore in the revised abstract), while at the same time better express the global inland water C cycle context.

**Section introduction: Lines 49–144 (clean version) and Lines 59–178 (revised version)**

***Ref: Sampling Design:*** *I think the selection of sampling sites and the sampling design needs more clarification and elaboration. How were inflowing rivers chosen and why were some sites sampled at a higher frequency? How were the lake sampling sites chosen, and importantly what depth was water sampled from? Lake sampling is not really described at all. How far downstream of the lake outlet was the outflowing river sampled? It seems to me the marsh sampling should be discussed in the same section as the water sampling, or at least before the analytical methods. Why were only marsh plants sampled and not terrestrial plants or phytoplankton, which are also probably important OM sources? The point of the cores from the marshes is not that clear- maybe explain a bit more the rationale for collecting cores as opposed to surface samples, and what information is provided by the depth profiles.*

Reply: We have added details on sampling in the revised version. Briefly, inflowing rivers were sampled along the entire lake area (when there was sufficient water flow), with a selection made for regular sampling based on their accessibility for collaborators on site and that these would have flow year-round. Lake sampling took place along a N-S gradient covered by boat, all water samples were taken with a Niskin bottle from the surface (approx 0.5 m depth).
**Section 2.2.2 Lake water sampling: Lines 230–233 (clean version) Lines 287–290 (revised version)**

***Ref: Analytical Methods:*** *This section is very dense and hard to follow. I would suggest splitting into subsections if journal allows. Some reorganization is needed. PCO2 measurements should proceed the estimates of DIC concentration since they are used for that. There is no mention at all of the CH4 measurements that are presented, which is a major oversight.*

Reply: The Methods section has been restructured as suggested, and CH$_4$ methodology has been added.
**Section 2.3.1 CH4 measurement and analysis: Lines 254–264 (clean version) and Lines 313–323 (revised version)**

***Ref: Results:*** *This section is very dense and hard to follow. If possible to break into subsections that would make reading easier.*

Reply: As for the section on analytical methods, this section is now broken up into subsections as suggested.
**Section 3. Result: Lines 379–498 (clean version) and lines 447–577 (revised version)**

**Ref: Vegetation data:** *As mentioned above, having d13C values of local terrestrial plants and phytoplankton, or at least estimates, would really be helpful and complement the data shown in Table 3. The 13C values of subsoils are referenced but the actual numerical values should be summarized, and it is unclear why topsoils are not included. In line 448 there is discussion of a mixing model based on plant isotopic values, but this is quite unclear and hard to follow. More details are needed, and the relative abundance of different plant types should be specified, ideally in a table.*

> Reply: Samples of terrestrial vegetation were taken and reported in Razanamahandry et al. (2022, https://doi.org/10.5194/bg-19-3825-2022). We refer to this in the revised version (section 3.10). Topsoil data (from the same study) have also been cited in the same section. Regarding phytoplankton, we sampled the total suspended matter pool in the lake (as discussed in the next comment); but see no way of sampling pure phytoplankton in such settings. As discussed in the manuscript, suspended matter in lake surface water are proposed to have a substantial contribution by phytoplankton, the discussion on this has been expanded in line with the next comment/suggestion.
> **Section 3.10 OC content and $\delta^{13}$C-OC of marsh vegetation: Lines 487–498 (clean version) and Lines 566–576 (revised version)**

**Ref: Phytoplankton 13C calculation:** *I'm afraid this is a gross oversimplification. This fractionation factor can very hugely and is sensitive to biological and environmental variables like growth rate and light, as well as differences between taxa. Use of a single value of 20‰ is too simplistic, and based on my quick review is on the high end, as opposed to an average value. So I don't think this is appropriate. In addition the uncertainty in this estimate needs to be accounted for. There is an extensive literature on this, but see for example:*

*Burkhardt, Steffen, Ulf Riebesell, and Ingrid Zondervan. "Effects of growth rate, CO2 concentration, and cell size on the stable carbon isotope fractionation in marine phytoplankton." Geochimica et Cosmochimica Acta 63.22 (1999): 3729-3741.*

*De Kluijver, A., Schoon, P. L., Downing, J. A., Schouten, S., & Middelburg, J. J. (2014). Stable carbon isotope biogeochemistry of lakes along a trophic gradient. Biogeosciences, 11(22), 6265-6276.*

*The Fry et al citation is quite outdated. Including more uncertainty in this calculation is required, and likely has effects on the inferred contribution of phytoplankton.*

> Reply: We fully agree that our approach is a simplification, and indeed there is quite some literature on factors affecting isotope fractionation in phytoplankton. Many of these are focused on marine phytoplankton, and while some principles will similarly apply to any type of aquatic environment, the lack of information on the phytoplankton community composition in our system provides a certain constraint. In the revised version we have updated

and expanded the discussion on this, citing the papers above as well as Lammers et al. (2017) and Bade et al. (2006).

Bade, D.L., Pace, M.L., Cole, J.J., and Carpenter, S.R.: Can algal photosynthetic inorganic carbon isotope fractionation be predicted in lakes using existing models?, Aquat. Sci., 68, 142-153, doi: 10.1007/s00027-006-0818-5, 2006.

Lammers, J.M., Reichart, G.J., and Middelburg, J.J.: Seasonal variability in phytoplankton stable carbon isotope ratios and bacterial carbon sources in a shallow Dutch lake, Limnol. Oceanogr., 62, 2773-2787, doi: 10.1002/lno.10605, 2017a.

Regarding the comment that the 20‰ value is on the high end: note that we used this value for isotope fractionation between phytoplankton and DIC, while most papers dealing specifically with variation of isotope fractionation in phytoplankton calculate their epsilon ($\varepsilon$) values between phytoplankton and $CO_2$. Given the difference of about 9 ‰ between $\delta^{13}C$ values of DIC and $CO_2$ (under pH and temperature conditions); this confusion likely explains why our initial value was considered high.

**Section discussion: Lines 563–576 (clean version) and Lines 647–663 (revised version)**

_Ref: **Consideration of hydrodynamic processes:** Differences in %POC between the TSM and soils was suggested to indicate a different, more organic rich source. However, this could also reflect sorting and selective settling of eroded soil that could lead to greater %POC in the lake. For example, soil minerals may selectively settle and deposit in sediments, whereas OM rich material may be more likely to be suspended. I think this is worth consideration. This could also potentially explain the higher %POC in the lake vs the inflowing rivers._

Reply: We have considered this aspect and added some new references (Bouchez et al. 2014, Repasch et al. 2022). There, however, is very limited empirical data on this from river systems, but in our interpretation, we would expect that the fraction ultimately transported to the lake itself would be the finer clay fraction – which typically has lower %OC, consistent with Repasch et al. 2022. Without more data on densities and %OC across different fractions of suspended sediments in our study area, it's not really possible to predict, but we have added a discussion on this in Section 4.1.
**Section Discussion: Lines 513–519 (clean version) and Lines 592–597 (revised version)**

_Ref: **More detailed implications:** I think it would be really valuable to see a bit more discussion of the implications of this work at the end of the discussion. What does this mean for the role of lakes in riverine carbon transport and emissions? Does the presence_

*of lakes in (tropical) rivers lead to a net increase or decrease in emissions, and does it change the overall source of C that is being respired? Do your results have implications for the carbon cycle effects of current anthropogenic changes in the catchment and the lake? Will greater erosion and/or rice production lead to changes in the GHG fluxes from this system or the downstream export of carbon? I think addressing some of these questions will enhance the impact of the paper.*

> Reply: This suggestion is in line with earlier suggestions on the introduction (context & objectives). We came back to this in the revised Discussion (see **section 4.2**). Note that some of the questions raised can only be addressed broadly and might be somewhat speculative, but we see the use of raising the questions and providing some pointers. We refer to a recent synthesis by our group (Borges et al. 2022, https://doi.org/10.1126/sciadv.abi8716) that showed that emissions from lentic systems were marginal to lotic systems at the scale of the African continent. The same seems to hold from the Amazon basin (Chiriboga et al. 2023, https://doi.org/10.1007/s00027-023-01039-6). Both studies are based on field measurements upscaled at larger scales with GIS so provide quantitative estimates albeit admittedly uncertain.
> **Section Discussion: Lines 641–652 (clean version) and Lines 733–744 (revised version)**

_Ref:_ *Line by line comments:*

*Line 38: this phrase about degassing is not clear to me.*

> Reply: This is reformulated as: "Nevertheless, the $pCO_2$ levels in the surface waters of the lake were lower than those in the inflowing and outflowing rivers. This reduction is likely due to the combined effects of phytoplankton production, which assimilates $CO_2$ during photosynthesis, and degassing processes. When $CO_2$-supersaturated riverine water enters the open lake, increased turbulence caused by wind fetch enhances gas exchange with the atmosphere, allowing $CO_2$ to escape more readily from the water column".
> **Section Abstract: Lines 37-41 (clean version) and Lines 44–49 (revised version)**

_Ref:_ *L61: This sentence is not clear. If the C fixed is not buried or emitted where does it go? Exported as DOC downstream?*

> Reply: This has been corrected as: "The study by Morana et al. (2022) showed that in situ primary production in some of the studied lakes could be ~20 times higher than the organic carbon (C) burial in sediments. This indicates that much of the carbon fixed through phytoplankton production is rapidly

recycled within the lake system, through processes such as microbial respiration, grazing, and remineralization, rather than being buried in sediments or emitted as $CO_2$ to the atmosphere. This finding challenges the traditional view of lakes as net heterotrophic systems (Del Giorgio et al., 1993; Duarte and Prairie, 2005; Aufdenkampe et al., 2011)".
**Section Introduction: Lines 63–68 (clean version) and Lines 73–79 (revised version)**

*Ref:* L70: *Be more clear why data from Madagascar is valuable in a global sense.*

Reply: We expand this line: "Madagascar's lakes, are unique as they lie within a region that combines tropical, subtropical, and anthropogenically influenced landscapes. These ecosystems are highly sensitive to environmental changes, such as deforestation, agricultural expansion, and climate variability, which affect carbon cycling processes and sediment fluxes. Understanding OC dynamics in Madagascar's lakes provides critical insights into carbon processing in subtropical regions with similar environmental pressures. This lack of data and uncertainty, especially in underrepresented regions like Madagascar, requires the collection of additional datasets over adequate spatial and temporal scales to refine global models of carbon cycling and better predict responses to anthropogenic and climatic changes."
**Section Introduction: Lines 78–85 (clean version) and Lines 93–100 (revised version)**

*Ref:* L90: *Specify how the wetlands have been altered.*

Reply:  We have specified this alteration of wetlands by adding lines the introduction section: "The wetlands surrounding the lake have undergone significant alterations, primarily due to their conversion into rice paddies to support local agriculture. This process involves draining natural wetlands, altering their hydrology, and replacing native vegetation with cultivated rice crops. These changes have profoundly impacted the wetlands' ecological balance, biodiversity, and natural functions (such as water filtration, flood control, and providing critical habitats for biodiversity) (Lammers et al., 2015). This transformation has been accompanied by extensive deforestation along the hillslopes, which has increased sediment runoff. Furthermore, the construction of channels and dams for irrigation in the floodplains has disrupted natural hydrological processes, further modifying the state of the Lake Alaotra wetlands."
**Section Introduction: Lines 101–109 (clean version) and Lines 121–129 (revised version)**

*Ref: L105: This is an important point and isn't totally clear here. Based on the Broothaerts paper there is a huge amount of sedimentation in the floodplain (100x greater than the lake) and a lot in the wetland (10x greater than the lake), leading to very low sedimentation in the lake itself.*

Reply: We expanded this section: "Studies on pollen from Lake Alaotra sediment archives have shown sedimentation rates of 0.3–0.6 mm y$^{-1}$ (Broothaerts et al., 2022), which are remarkably low given the high erosion rates in the catchment. This apparent discrepancy can be explained by the significant trapping of sediments in the floodplain and wetlands surrounding the lake. According to Broothaerts et al. (2022), sedimentation rates in the floodplain are approximately 100 times greater than in the lake, while wetlands exhibit rates about 10 times higher. These findings suggest that the majority of sediments mobilized from hillslope erosion are deposited in upstream floodplain and wetland areas, leaving relatively little to reach the lake itself.

Furthermore, sedimentation rates in the lake have remained consistently low over the last 1000 years, with no significant increase observed despite extensive land-use changes in the catchment (Broothaerts et al., 2022). This highlights the critical role of the floodplain and wetlands as sediment sinks, buffering the lake from excessive sedimentation. This process is further explored in the discussion and in a follow-up paper that examines elemental and stable isotope data from floodplain, marsh, and lake sediment cores, which reinforce these observations."

**Section Introduction. Lines 123–135 (clean version) and Lines 148–160 (revised version)**

*Ref: L130: This sentence is redundant with the earlier part of the paragraph.*

Reply: These sentences have been merged: "The region experiences a tropical climate with two distinct seasons: a hot, rainy season from November to April and a cooler, dry season from May to October, which aligns with the division of our sampling period into (1) a dry season (June to October), characterized by cooler temperatures and lower precipitation, and (2) a rainy season (November to May), with higher temperatures and increased rainfall (Supplementary Figure S1)"

**Section 2.1 study area: Lines 153– 160 (clean version) and Lines 188–195 (revised version)**

*Ref: Figure 2: Is it possible to add how this hydrological difference affects lake level?*

Reply: We were unable to find historical lake level data; otherwise see Figure 3 where water levels for the outflowing river (Maningory) are shown for 2018-

2019; this should be a reasonable proxy for lake water levels (and nicely matches the longer-term outlet discharge in terms of timing and shape).
**Figure 3: Lines 385 (clear version) and Lines 453 (revised version)**

*Ref: L176: Awkward phrasing- try to rewrite.*

Reply: Point taken, this sentence has been rephrased: "The expansion of rice fields in the areas surrounding the lake has predominantly occurred through the conversion of wetland ecosystems (Mietton et al., 2018).
**Section 2.1: Lines 203–204 (clean version) and Lines 257–259 (revised version)**

*Ref: L195: Much more details on lake sampling needed.*

Reply: More details has been added in the revised version.
**Section 2.2.2 Lake water sampling: Lines 229–233 (clean version) and Lines 287–291 (revised version)**

*Ref: L298: There is not really much on d15N in the paper. Was it used at all? I recognize the authors are providing a wide range of data that was not used in the paper. Perhaps methods for these analyses should be in a supplement so that they can be used later but do not distract in the main text.*

Reply: Indeed, we found no clear use for them in this manuscript but feel it important to put the $\delta^{15}N$ data out (as well as some other parameters), as they might still prove to be useful for other researchers e.g. for data syntheses or meta-analyses. We modified the Methods to explicitly mention the methodology (measured in the same run along with OC, PN, and $\delta^{13}C$): $\delta^{15}N$ data were measured alongside POC, PN, and $\delta^{13}C$, and while they are not discussed in detail in this study, the full data are provided in the supplementary material to ensure the data could be used in future research.
**Section 2.3 Field and laboratory analyses: Lines 291–292 (clean version) and Lines 351–352 (revised version)**

*Ref: Figure 6: The cause of the gaps in data are not clear to me, make clearer here or in the methods.*

Reply: $pCO_2$ was measured in situ (LICOR-820, headspace equilibration) during our own field campaigns, but not by the team performing the regular monitoring for logistical reasons. $CH_4$, in contrast, was analyzed on discrete samples preserved on site and measured later in the lab – hence, for this parameter a more complete dataset is available. For logistical reasons, not all of the rivers sampled during our own field campaigns were monitored

regularly, this was only the case for a sub-set of rivers. This explanation has been added into the text.
**Section 2.3 Field and laboratory analyses: Lines 254–276 (clean version) and Lines 313–335 (revised version)**

*Ref: L358: It is interesting that %POC is lower in the outflow than the lake. What is the source of inorganic TSM in the outflow?*

Reply: This is an interesting point. Note however, that the Maningory data (outflow) include data from regular sampling throughout the year, while the lake samples were taken during a short time periods of the full field campaigns; given that there is no real overlap in sampling dates we cannot perform a paired comparison. The $CH_4$ data, however, indicate lateral inputs from riparian wetlands to the Maningory, which might be one possible hypothesis. The sampling site of the lake outflow is located at the road bridge, which is about 4-5 km downstream of the point where water flows out of the lake, and some rice fields are located in the area in between; hence drainage from these might deliver more mineral-rich inputs. No changes have been made in the manuscript. We have noted this in the revised Methods section.
**Section 2.2 Sampling design: Lines 226–228 (clean version) and Lines 282–286 (revised version)**

*Ref: Figure 9: Is any information on the age-depth relationships in the cores available. The core data in general is not that informative, so maybe it could be more simply summarized as a source of OM.*

Reply: Yes, [14]C dating was performed on the M3 core, these are reported in Broothaerts et al. (2022). It was suggested elsewhere to summarize the core data, which we decided to do; hence we will discuss depth trends (and the dating) in a subsequent paper along with data on sediment cores from the floodplains and lake Alaotra.

*Ref: L454: Give numerical values (i.e. average plus standard deviation) for the lakes in East Africa being compared to.*

Reply: These data have been summarized and added in the revised version:

"The mean daily primary production rates we measured (0.5 ± 0.3 g C m-2 day-1) were moderate compared to the broader range of East African lakes, where primary production averages around 10 ± 5 g C m$^{-2}$ day$^{-1}$ (Morana et al., 2022), and similar to e.g. those measured in the oligotrophic lake Kivu (0.6 g C m-2 day-1; Darchambeau et al., 2014)"
**Section Discussion: Lines 548–551 (clean version) and Lines 632-636 (revised version)**

*Ref: L469: In addition to the issues discussed above, if there is high C fixation this can lead to enrichment of d13C in the water column (i.e. a Rayleigh distillation effect) and potentially lead to erroneous estimates based on the fractionation factor.*

*See for example: Van Dam, Bryce R., et al. "CO2 limited conditions favor cyanobacteria in a hypereutrophic lake: an empirical and theoretical stable isotope study." Limnology and Oceanography 63.4 (2018): 1643-1659.*

> Reply: In line with the earlier comment on isotope fractionation for phytoplankton, we re-examined this aspect of the study and included a more critical discussion of factors influencing isotope fractionation between DIC and phyto biomass.
> **Section 4 Discussion: Lines 563–576 (clean version) and Lines 647–662 (revised version)**

*Ref: L474: This begs the question: are the differences between the inflow water and the lake/outflow water significant?*

> Reply: Yes, indeed – for some reason we have not mentioned this; we now mention this specifically.
> **Section Discussion: Lines 509–510 (clean version) and Lines 588–589 (revised version)**

*Ref: L513: Again, provide numerical values for these other lakes.*

> Reply: Values have been included in the revised version.
> **Section Discussion: Lines 624–627 (clean version) and Lines 712–713 (revised version)**

*Ref: L517: Need a citation for these data from the Congo.*

> Reply: Reference has been added: (Borges et al, 2022)
> **Section discussion: Lines 628–631 (clean version) and Lines 719–723 (revised version)**

*Ref: L524: I think a more detailed explanation for the connectivity causing high pCO2 and pCH4 is needed.*

> Reply: This section has been expended. Section 4.2 Sources of DOC in the Lake Alaotra system
> **Section Discussion: Lines 641–652 (clean version) and Lines 733–744 (revised version)**

**RC2**: Dailson Bertassoli, 16 Oct 2024

*Ref: General comments:*

*This study investigates the carbon biogeochemistry of Lake Alaotra, Madagascar, by analyzing variations in carbon pools, $CO_2$, $CH_4$, and other parameters over a complete hydrological cycle. It offers valuable and much-needed data that contribute to advancing discussions on the role of tropical lakes as "biogeochemical reactors." While the authors have made an important effort to underpin their discussion, I believe that some broad generalizations weaken certain key findings of the research. Therefore, I believe this manuscript should undergo major revision or be resubmitted before publication.*

*The introduction offers extensive details about the study area but lacks sufficient emphasis on the main research objectives, making it difficult to fully understand the rationale of the study. The authors should reconsider the level of attention given to 'lavakas' (l12, l83–l86, l124) and paleoenvironmental interpretations (l108–112), as these topics are not directly connected to their main findings. Additionally, the manuscript would benefit from clarifying the gaps in the carbon cycle that this research aims to address, providing a stronger rationale for the study.*

> Reply: This suggestion is in line with those of Ref#1, see response there – we re-organized parts of the Abstract and Introduction (and Discussion) to frame the study more in the context of global/regional C cycling and the role of (sub)tropical lakes. The paleo-environmental proxy context was and remains an important justification for us to have conducted this work, but we removed it from the abstract, and have tried (in the introduction and Discussion) to express this better and place it more in the background and link to published and submitted companion papers; we understand the link might not be obvious in the original manuscript.
> **Section Abstract: Lines 12–47 (clean version) and Lines 12–57 (revised version)**
> **Section Introduction: Lines 49–144 (clean version) and Lines 59–169 (revised version)**

*Ref: The authors mention a "selection of rivers," but the criteria for choosing these two rivers and their representativeness regarding the overall water balance of Lake Alaotra are not clearly explained. Additionally, highlighting these rivers on the map in Figure 1 would improve visualization. There is also some ambiguity about how many rivers were sampled during the high- and low-water field trips. Overall, the sampling scheme and the methodologies used are somewhat unclear and lack key details. For example, what were the water depths at the sampling sites? How were the sampling points in the lake selected? While coordinates are provided in the Supplementary Data, important implications regarding the sampling strategy should be discussed in the main text. For instance, $CH_4$*

*concentrations in lakes can vary significantly depending on proximity to the margins. Addressing these aspects would greatly enhance the study.*

Reply: These are mostly in line with suggestions from Ref#1 and have been taken into account: more details have been added in the Materials and Methods section, and rivers where sampling was done regularly are now indicated with a separate symbol on the revised Figure 1.
**Figure 1: Lines 179 (clean version) and Lines 230 (revised version)**

*Ref: The results section would be much more reader-friendly with the inclusion of additional graphs and tables. Biplots, in particular, could greatly aid in comparing the ranges of organic carbon across soil, lake, and river samples. In the discussion, although the evidence suggesting that the increase in %POC in the lake is likely driven by phytoplankton input is relatively solid, the text is structured in a way that makes the argument somewhat unclear. A clearer and more focused presentation of this idea would improve the coherence of the discussion. Also, the authors may get interesting perspectives on erosional patterns by delving deeper into the TSM changes.*

Reply: We have added a new Figure (Fig. 10) which plots $\delta^{13}$C values versus %POC for different relative end-members (forest soils, grassland soils; data from Razanamahandry et al. 2022; marsh vegetation and peat cores) versus and of suspended matter in the inflowing rivers, Lake Alaotra, and the lake outflow. The corresponding discussion in the text should contribute visually to the revised discussion on phytoplankton inputs.
**Figure 10: Lines 580 (clean version) and Lines 668 (revised version)**

*Ref: My main concern, however, lies in section 4.2. It is important to emphasize that the separation between POC and DOC fractions, based on size, represents an "instrumentalist" approach that overlooks key factors. Most importantly, a primary control on %POC and TSM, the energy of the environment, was severely neglected in the discussion. This, naturally, has significant implications for some of the interpretations. Similarly, degradation, which is also size-dependent, probably play a critical role in shaping the OM δ13C signatures observed in the system but was not adequately addressed. Additionally, although the conclusion that DOC and POC sources are uncoupled is reasonable, the entire discussion regarding DOC sources seems to oversimplify the system and is not fully supported by the data presented in the article.*

Reply: The distinction between POC and DOC is indeed operationally defined. We assume the comment on the energy of the environment corresponds to the comment of Ref#1 regarding hydrodynamics influencing particle composition (deposition, resuspension, particle sorting, etc.) -see our reply there. Regarding degradation, this may indeed have small effects on $\delta^{13}$C although we do not see that this will fundamentally change our conclusions; keeping in mind for example that potential carbon sources discussed (e.g. grassland and forest soils) are already composed of substantially decomposed

organic matter; changes in $\delta^{13}C$ upon decomposition typically are most prominent in the initial phase. We added a reference to Geeraert et al. (2016) regaring potential shifts in $\delta^{13}C$ of DOC during early stages of decomposition; here too, however we do not see a major influence on our interpretations. We are unsure what the reviewer refers to with the statement that the discussion on DOC sources is not fully supported by the data presented – some more details would have been welcome. In line with other feedback on the $\delta^{13}C$-phytoplankton discussion, we have modified the corresponding discussion for phytoplankton-derived DOC, and have made several minor clarifications in section 4.2.

Geeraert N, Omengo FO, Govers G, & Bouillon S (2016). Dissolved organic carbon lability and stable isotope shifts during microbial decomposition in a tropical river system. Biogeosciences 13: 517-525

**Section Discussion: Lines 614–616 (clean version) and Lines 703–705 (revised version)**

*Ref: Lastly, it would be great if the authors could place the obtained results within a broader regional context, highlighting their implications for the current understanding of the tropical carbon biogeochemistry cycle.*

Reply: This suggestion is in line with those of Ref#1, see response there – we removed the paleo-environmental reconstruction context from the Abstract (but retain a mention of it in the Introduction and Discussion), and have rewritten parts of the Introduction and Discussion to frame the study more in the context of global/regional C cycling and the role of (sub)tropical lakes. The paleo-environmental proxy context was and remains an important justification for us to have conducted this work as well as for submitted or published companion paper, but we understand this is perhaps less evident without actually presenting and discussing sediment core data.
**Section Introduction: Lines 133–144 (clean version) and Lines 158–169 (revised version)**

*Ref: Specific comments:*

*L12-14: As I read these lines, I thought the main focus of the article to be different. Consider focusing the beginning of the abstract towards the main target of this article.*

Reply: See response to previous comment.
**Section Abstract: Lines 12-15 (clean version) and Lines 12–16 (revised version)**

*Ref:* L26: Not necessarily 'surprising'

> Reply: It was surprising to us. The vast majority of lakes we have studied so far show little or no change in DOC concentrations (and $\delta^{13}C$) from the inflowing rivers. See Discussion on lines 504-517 in the initial version of the manuscript. Unless this is considered a critical point, we would prefer to leave this as it is.

*Ref:* L31: I'm not sure if "was expected" is the right expression here. Do you mean "was found," as in: "δ13C data indicated that marsh vegetation was the main source of net DOC inputs, while phytoplankton likely contributed to POC in the lacustrine waters."?

> Reply: We rephrased this line to: "However, sources of POC and DOC were uncoupled: $\delta^{13}C$ data indicated that marsh vegetation was the main source of net DOC inputs, while phytoplankton likely contributed substantially to POC in the lacustrine waters, at least during some of the sampling period."
> **Section abstract: Lines 30–32 (clean version) and Lines 35–39 (revised version)**

*Ref:* L100: I recognize that this study is present in the literature, but its conclusions don't seem well-supported by the findings. I'm not sure if continuing to reference it truly benefits the advancement of science.

> Reply: Fair point – we indeed agree that their conclusions are not supported by subsequent studies (and we feel this is clear from the rest of the sentence and the following one). We have merged both sentenced and shortened it, so that it should be more clear now that this is a disputed claim – we still feel this is worth mentioning (and explicating that newer data suggest the claim was incorrect). **Section introduction: Lines 119–122 (clean version) and Lines 142–147 (revised version)**

*Ref:* L108-L112: I am not sure how this directly relates to the main subject of the article.

> Reply: We feel this is an important justification or aspect of the context of this paper: the lake is situated in a region that has experienced major anthropogenic disturbance and deforestation, and as stated in the introduction one of our broader project aims is also to use lake sedimentary records to reconstruct some of these changes. We would thus prefer to keep this sentence.

*Ref:* Figure 1: Please consider highlighting which rivers were measured monthly and which were not measured during the dry season.

Reply: Figure 1 has been fully updated, and the distinction between rivers sampled on regular intervals (monitoring) and those only sampled during intensive field campaigns is now clear.
**Figure 1: Lines 179 (clean version) and Lines 230 (revised version)**

*Ref: Figure 9: Are there any considerations regarding changes in the parameters for different depths that can contribute to your discussion?*

Reply: We acknowledge that the data in Figure 9 (of the original version) are currently not discussed in detail: that will be part of a broader manuscript in preparation that discusses similar data from across the floodplain-marsh-lake gradient; and we mainly presented them here since they show %OC and $\delta^{13}$C values from what we consider to be a potentially important 'end-member' source of OC to the lake. Ref#1 suggested to replace this Figure with a summary, rather than present the full profile, and we have decided to follow their suggestion (data now summarized in Table 3).
**Table 3: Lines 486 (clean version) and Lines 565 (revised version)**

---

## Author Response (AR3)

**REPLIES TO TECHNICAL CORRECTION**

*Dear Editor,*

Thank you for your feedback and for considering our revised manuscript for publication. Below are our responses to the requested technical corrections:

*-Title: Please remove the period at the end.*

Reply:

We have removed the period at the end. **Line 3 (Corrected Manuscript)**

*-Line 18 (track-change version) "stable isotope C ratios": A more proper phrase would be "stable C isotope ratios" or "stable isotope ratios of C". Or C can be removed because C repeats in the following words.*

Reply:

We have removed C for better clarity. **Line 18 (Corrected Manuscript)**

*- L 43: flooded "forests"*

Reply:

We have corrected "flooded forests" as suggested. **Line 43 (Corrected Manuscript)**

*- L 576 "16 ‰.": Please remove the period following per mil.*

Reply:

The period after "16 ‰" marks the end of the sentence. **Line 567 to 570 (Corrected Manuscript)**

*- L 670 "Our data-results": "Our results" would be better.*

Reply:

"Our data-results" has been changed to "Our results" for clarity. **Line 659 (Corrected Manuscript)**

*- Figure 1 legend: Please check whether the text reflects changes in legend colors.*

Reply:

We have checked and ensured that the text correctly reflects the changes in legend colors.

**Line 181 to 183 (Corrected Manuscript)**

*- Figure 10 legend "Phytoplankton (estimated ranges)": If this is based on a citation, "estimated ranges" could be replaced by the reference.*

Reply:

We now refer to the text, where more information can be found on how these estimates were derived and which references we based them on: "(see text for details on how the latter were derived)". **Line 581 (Corrected Manuscript)**